# Automatic detection of methane emissions in multispectral satellite imagery using a vision transformer

Bertrand Rouet-Leduc [1,2,3] ✉ & Claudia Hulbert[2,3]

Curbing methane emissions is among the most effective actions that can be taken to slow down global warming. However, monitoring emissions remains challenging, as detection methods have a limited quantification completeness due to trade-offs that have to be made between coverage, resolution, and detection accuracy. Here we show that deep learning can overcome the trade-off in terms of spectral resolution that comes with multi-spectral satellite data, resulting in a methane detection tool with global coverage and high temporal and spatial resolution. We compare our detections with airborne methane measurement campaigns, which suggests that our method can detect methane point sources in Sentinel-2 data down to plumes of 0.01 km$^2$, corresponding to 200 to 300 kg CH$_4$ h$^{-1}$ sources. Our model shows an order of magnitude improvement over the state-of-the-art, providing a significant step towards the automated, high resolution detection of methane emissions at a global scale, every few days.

Methane is the second largest contributor to global warming, estimated to account for approximately a third of warming to date[1,2]. The strong contribution of methane to global radiative forcing, in combination with its much shorter atmospheric half-life compared to CO$_2$, makes the reduction of methane emissions critical in the short-term fight against climate change. Yet, despite recent efforts and regulations introduced around the globe to limit emissions, atmospheric methane levels are steadily increasing and recently reached an all time high[3].

Methane emissions can be intermittent or persistent in time, with a small fraction of large sources contributing disproportionally to total emitted volumes[4]. The systematic identification and quantification of such sources is key to prioritizing and validating mitigation actions, and to building precise methane inventories. Governments and organizations rely on limited information to introduce remedial actions, as current detection approaches and inventories suffer from severe drawbacks. Bottom-up inventories have been shown to under-estimate emissions, often dramatically[2,5–7]. Current methane measurements are limited in scale and/or resolution: detectors mounted on ground vehicle and plane or drone have limited coverage, while hyperspectral satellites suffer either from very poor resolution or from limited coverage and the need for tasking[8].

Satellite-based methane detection generally relies on identifying absorption in the short-wave infrared (SWIR) of the backscattered sunlight, in spectral regions known to be absorbed by methane. Hyperspectral satellites can provide a high SWIR spectral resolution that enables the precise determination of methane column concentration data (XCH$_4$)[9–13] (which can be further refined using machine-learning methods[14–16]). However, hyperspectral satellites trade this high spectral resolution for either low spatial coverage (e.g., target-mode satellites PRISMA) or low spatial resolution (e.g., Sentinel 5, with detection capabilities on the order of several tons/h), thereby providing limited quantification completeness. In order to circumvent these limitations, there has been a growing interest in developing methane detection techniques in data from multi-spectral satellites, such as ESA's Sentinel-2 constellation, that scans the entire Earth every 2 to 5 days with high spatial resolution (20 m in SWIR bands). However, multi-spectral satellites make the opposite trade-off, providing high spatial and temporal resolution along with global coverage at the cost of much less spectral information (with e.g., a dozen spectral bands for

[1]Disaster Prevention Research Institute, Kyoto University, Japan. [2]Geolabe, Los Alamos, NM, USA. [3]These authors contributed equally: Bertrand Rouet-Leduc, Claudia Hulbert. ✉e-mail: rouetleduc.bertrand.5s@kyoto-u.ac.jp

Sentinel-2, versus hundreds for hyperspectral satellites). Consequently, methane detections in multi-spectral data tend to be very noisy, and so far have only provided the ability to detect large emissions[17,18], down to 2 to 3 tons/h over very bright surfaces such as deserts[8] (e.g., Turkmenistan or southern Algeria) and 10+ tons/h in non-optimal conditions (e.g., the Permian Basin in the U.S.)[18].

Here we developed a deep learning architecture tailored for existing open-source multi-spectral satellite data, with the goal of automatically identifying methane signatures and deconvolving signal from noise. We find that our approach drastically improves methane detection capabilities, enabling to detect emissions one order of magnitude smaller than the state-of-the-art on the same data. Our results suggest that our model can detect methane emissions down to plumes of 0.01 km², corresponding to methane leak rates of 200 to 300 kg/h (with variations depending on wind conditions). Leaks of this size account for the vast majority of the estimated methane budget coming from point sources for most airborne campaigns in California, Colorado, and the Permian Basin analyzed in a recent survey[19]. Our results also show that our model can detect all the methane releases that have been timed with Sentinel-2 overpasses down to 1100 kg/h, with a gap in S2 controlled releases below that threshold that we hope will be filled in future tests of our method. Our approach represents a significant step towards the automated monitoring of persistent methane emissions, anywhere on Earth, every few days, and would require few modifications to enhance detection capabilities in other multi- and hyper-spectral constellations.

## Results

### Learning to detect synthetic CH₄ plumes in Sentinel-2 data

Deep learning requires large amounts of training data, but limited ground truth exists for methane detection in multi-spectral images. We therefore rely on synthetic methane plumes instead, in an approach conceptually similar to our previous work aiming at extracting small ground deformation signals from radar satellite data[20] and at extracting small elasto-gravity signals from seismic data[21].

To this end, we gathered a large database of pairs of Sentinel-2 tiles, sampled at two consecutive times with limited cloud cover (less than 25%) in a number of regions representing a variety of different climates, topographies, and land uses. This database contains a total of 900 pairs of Sentinel-2 tiles (about 10.8 million km²), from areas that tentatively do not encompass known potential methane sources (such as oil & gas fields, landfills, etc.). The data is cut into 2.5 × 2.5 km² scenes, resulting in about 1,650,000 unique Sentinel-2 samples at two consecutive acquisition times. The Sentinel-2 data consists in L1C Top-of-Atmosphere (TOA) reflectance from Sentinel-2A and 2B. These images are made of 13 different spectral bands with a square pixel resolution varying from 10 to 60 meters. The input bands are re-sampled to the 20 m resolution of bands 11 and 12.

In order to build a training dataset of Sentinel-2 satellite images containing synthetic methane leaks, we assume the leaks stem from point sources. Instead of physically realistic WRF-LES simulation schemes[22], we opted for simple Gaussian plumes to efficiently generate thousands of different patterns, with the goal of creating a diverse training dataset without attempting to simulate detailed physical processes (which tend to be lost in the noise after embedding in S2 data, especially for small leaks). We generated thousands of Gaussian plumes with various emission rates and wind velocities, to which we added auto-correlated atmospheric noise to mimic atmospheric turbulence. The plumes were then embedded into about half the Sentinel-2 2.5 × 2.5 km² scenes using the Beer-Lambert law[18], at a random location. More details on the training, validation, and testing data, and on the synthetic plume generation and embedding, can be found in the *Methods* section below.

We train an encoder-decoder architecture in order to identify the synthetic methane plumes embedded into real Sentinel-2 images, with a transformer encoder. Initially introduced for machine translation, transformer models[23] have achieved state-of-the-art performance in natural language processing[23], computer vision[24–26], and audio processing[27,28]. The main innovation introduced by early transformers architectures consisted in including self-attention modules in encoding and decoding blocks, i.e., mechanisms used in sequence modeling that allow to model dependencies within sequences irrespective of distance[29,30] in the data. In comparison with convolutional networks, self-attention introduces the ability to capture long distance interactions, and improves training efficiency[23,31].

Besides a transformer architecture for the encoder, our model has a classic U-net architecture[32,33], that connects the encoder to the decoder at different resolutions. The model learns to recognize methane signals as a sequence-to-sequence prediction problem, which delimits the extent of the detected methane plume. The input to the model is a stack of bands B1, B2, B3, B4, B5, B8, B8A, B9, B11 and B12 at two times $t-1$ and $t$. The task of the neural network is to find the methane leak in the input images, defined as classifying the set of pixels corresponding to the embedded synthetic plume at time $t$. A schematic of our training approach is shown in Fig. 1.

### Performance on synthetic data

The 1,650,000 unique Sentinel-2 samples at two consecutive acquisition times are split in training, validation, and testing sets. About 75% of the data is used for training, and comes from tiles over Canada, Egypt, England, Ethiopia, France, India, Iran, Japan, Kenya, Mali, Mexico, Morocco, the US, and Saudi Arabia. About 10% of the data is used for validation, and comes from tiles over the south of Argentina, Belgium, and China, while about 15% of the data is finally used for testing, and comes from tiles over Afghanistan and the north of Argentina. The training, validation, and testing regions are different in order to improve the generalization of the model retained, which has the best performance on the validation set.

Figure 2a summarizes the results of evaluating our model on the testing set, in terms of F1 score (harmonic mean of precision and recall, see Methods for details) as a function of signal to noise ratio (SNR, defined as the ratio of the mean B12 reflectance reduction due to the synthetic methane plume to the standard deviation of B12 in the sample). These results show that our model can reliably detect methane in Sentinel-2 data down to about 5% SNR, about an order of magnitude improvement over using a threshold on the normalized Multi-band multi-pass method (MBMP, see Methods), the core component of state-of-the-art methods for methane detection in Sentinel-2 data[17,18]. Most of the synthetic emissions missed by our model comes from very small plumes at low SNRs (below 5%). We explain the strong performance of our model by three main factors: (i) relying on Gaussian plumes randomly embedded in real Sentinel-2 data (instead of real plumes or computer-intensive WRF-LES simulations), enables us to generate a training dataset that is orders of magnitude larger than the ones typically used in previous attempts at developing deep learning models for methane detection[16,34], thereby enabling to fully train large deep learning models. (ii) The use of two time-steps as input (conceptually similar to the MBMP approach) enables the model to use the first image as a reference image, to which the second image is compared in order to identify transient signals in methane absorbing band 12, while false positives in band 12 can be discriminated using the other bands and their evolution over the two time-steps. This comparison is crucial to distinguish signal from noise and correctly detect methane plumes, in particular the smaller ones. And (iii) the use of transformers instead of convolutional neural networks (CNNs) enables our model to capture the long-range nature of a plume.

Figure 2c shows the receiving-operator curves for classifying methane-containing Sentinel-2 pixels, which summarizes the true positive rate and false positive rate of our deep learning model and the MBMP method at different thresholds on their outputs. These results

show that our deep learning model can be used as a methane detector in Sentinel-2 data with an extremely low false positive rate (for example less than 0.03% pixel-wise false positive rate for 85% true positive rate for the threshold on the model's output used to make Fig. 2a, for all of the samples in the test set, whose SNR distribution is shown in Supplementary Fig. S1), while no classifier can be built solely from a threshold on the MBMP method that has a low false positive rate (at low SNR). We argue that a very low false positive rate is the key to move towards a truly automated methane detection method.

Figure 2b shows examples of our approach of embedding synthetic methane plumes in real Sentinel-2 data, and tasking a deep learning model with retrieving the associated pixels. These examples come from the test set and are made from Sentinel-2 data from regions not included in the training nor validation sets, while the synthetic plumes also come from a separate test set of plumes. Each column is a different example, that shows the band B12 data after plume embedding, the detection from the MBMP method and our deep learning method with the thresholds from Fig. 2b, and the ground truth. Except for the band B12 image, the detections and ground truth are overlaid on the natural colors RGB image from bands B2, B3 and B4. The thresholds that determine the classification of a pixel as containing methane are shown in Fig. 2b, and the same thresholds are used throughout the manuscript.

Importantly, we note that the false positive rate of our model is evaluated here on real and unaltered Sentinel-2 data. Only the positive samples of our databases have a synthetic element, while the negative samples are original Sentinel-2 data.

## Performance on real methane plumes

In order to evaluate our model on real methane leaks, we analyze the *Methane plumes from airborne surveys* open-source dataset published by Carbon Mapper[35]. This dataset contains methane plumes detected during several airborne surveys conducted between 2020 and 2021, performed with both the Global Airborne Observatory (GAO) and the

Next-Generation Airborne Visible/Infrared Imaging Spectrometer (AVIRIS-NG). The dataset contains a total of 2526 methane plumes, spanning leak rates from about 8 kg/h to about 9000 kg/h, with a median emission rate of about 240 kg/h. The airborne campaigns were conducted in a number of different locations (California, New Mexico, Utah, Texas, Colorado, Louisiana, and Pennsylvania), and thereby encompass a variety of different environments, ranging from deserts to humid and vegetated areas. Furthermore, methane plumes are detected for a variety of different sources (oil and gas infrastructures, coal mines, landfills, animal farming facilities, etc.).

By systematically applying our trained deep learning model to Sentinel-2 multi-spectral images close in time to cataloged airborne detections, we are able to build statistics evaluating the performance of our approach on a large number of real methane plumes. For each methane plume detected in the Carbon Mapper catalog, we crop a pair of $2.5 \times 2.5$ km$^2$ Sentinel-2 images that encompasses the plume's cataloged location, and feed it to our deep learning model. In each case the image is centered on the Carbon Mapper cataloged location, but importantly, in training the methane leaks are uniformly distributed in the $2.5 \times 2.5$ km$^2$ inputs, and therefore there is no a priori bias towards detection near the center of the input (as shown in Fig. S5 in the Supplementary Materials). This pair of Sentinel-2 images consists in a reference date in the 3 months prior to the leak, and a detection date in the 7 days prior to the leak. We further restrict our analysis to cloudless days (less than 0.5% cloud cover), which yields a total of 7724 possible pairs for the 2526 leaks of the catalog.

We note that a caveat is that plumes may be intermittent, and not present when our Sentinel-2 images are taken on a different day than the airborne detections. Previous studies[36] suggest that the persistence of leaks (defined in ref. 36 as the average fraction of airborne detection on different days for a given leak) in the catalog is around 20 to 26%. This average leak persistence is what we also expect to detect at best with a satellite-based method when attempting to detect the cataloged leaks on different days than the cataloged detection.

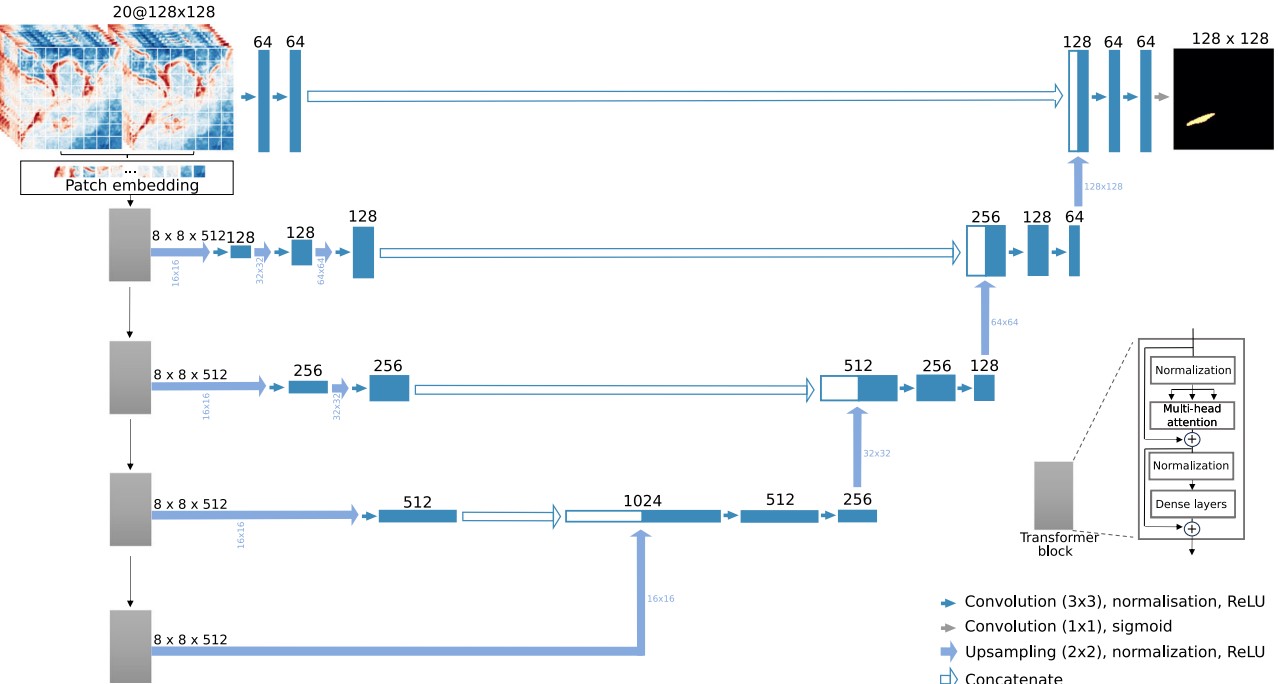

**Fig. 1 | Architecture of our methane detector neural network.** The model takes as input ten bands of Sentinel-2 multi-spectral data, at two different times $t-1$ and $t$ (20 channels input), and is tasked with finding the input pixels where a synthetic plume has been embedded, at time $t$, with time $t-1$ being used only as a reference. The model has an encoder-decoder architecture, where the encoder is a vision transformer (ViT)[26] whose blocks plug into the decoder of a U-Net[32] with matching up-sampling using deconvolutional layers. The input data is fed to the transformer encoder in the form of a 1D sequence of patches. The model is trained on 1,235,000 pairs of Sentinel-2 samples for ten epochs, and the model that has the best performance on the separate validation set is kept.

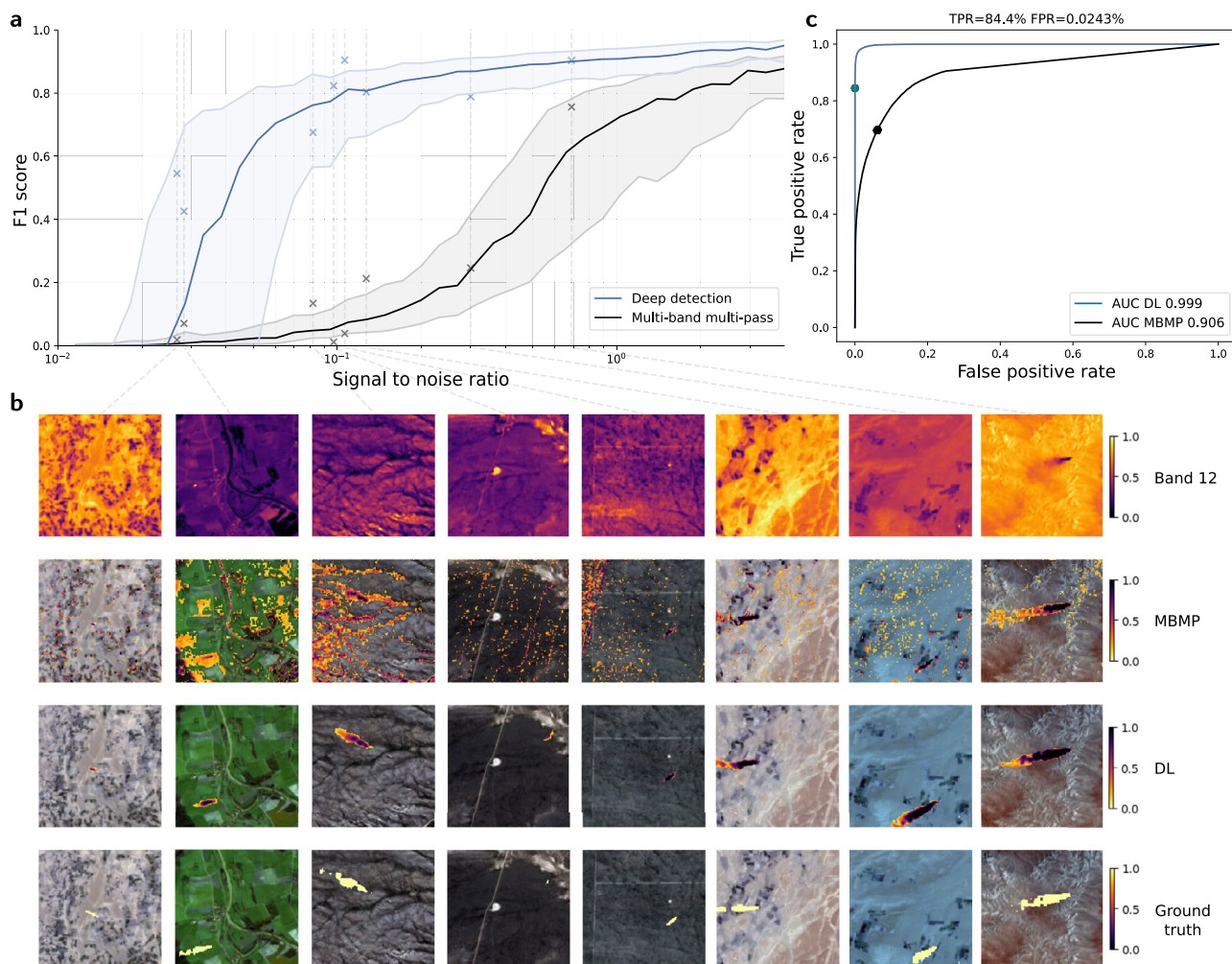

**Fig. 2 | Performance of our deep learning approach versus a multi-band-multi-pass (MBMP) approach. a** Performance in terms of F1 score (see Methods) of our deep learning model (in blue), compared with a normalized MBMP method[17] (in black), as a function of signal-to-noise-ratio (SNR). The thick lines are the median of 40 bins (of equal number of samples along the SNR axis), and the thin lines are the 25 and 75 percentiles. **b** Examples of Sentinel-2 data with embedded synthetic plume. Each column is an example at a different SNR, with the first row showing band 12 after plume embedding, the second row showing the MBMP method, the third row showing the application of our deep learning model, and the last row showing the ground truth of the plume's mask. The examples are placed in (**a**) according to SNR (following the gray dashed lines) and F1 scores of the MBMP (black crosses) and our method (blue crosses). The colorbars are the same for each row and correspond to the normalized band 12 for the first row, to the classifiers outputs for rows 2 and 3, and show the plume's mask for the last row. **c** Receiver-operator curve that shows the ratio of true positives to false positives for various classifying thresholds and for various signal to noise ratio conditions, with the thresholds used in (**a**, **b**) shown as dots. Note that false positives here are *not* synthetic, but come from real Sentinel-2 data, as only the added plumes are synthetic. The data shown here are from the test set (for about 40,000 samples), that comes from different regions than the regions used for training, and the synthetic plumes used in the testing set are different from the training set plumes.

Another caveat when evaluating methane detections in e.g., oil and gas sites is that it is to be expected that other leaks (in particular smaller ones, as leaks follow a power law[19]) are sometimes present within the same tile, and may not be cataloged, especially when analyzing data from different days than the airborne detections.

Figure 3 summarizes the results of our evaluation over seven days preceding each cataloged leak: for each pair of $2.5 \times 2.5\,km^2$ tiles centered on a cataloged plume's location and fed to our deep learning model, we consider the detection of the leak to be successful if our model detects methane for at two adjacent pixels within 500 m of the leak. Fig. 3 shows the fraction of the cataloged leaks detected by our method, as a function of leak rate. The Carbon Mapper catalog regroups data from campaigns performed with two instruments (AVIRIS-NG and GAO). Our detection results are summarized separately for the AVIRIS-NG (black) and GAO (gray) campaigns. For both instruments, each of the 10 bins shows the average detection rate of

our deep learning model and the average leak size and rate in the bin, for 332 and 442 pairs of Sentinel-2 images per bin, respectively.

Figure 3 also summarizes our detection rate when applying the exact same methodology in the absence of known leaks, for three different tests. (i) The blue bin shows our average detection rate when applying our model to pairs of Sentinel-2 images over southern New Mexico (but away from the Permian Basin), and provides an estimated false positive rate of 0.7% in conditions similar to that of the Permian Basin. (ii) The green bin shows our false detection rate using the exact same methodology as elsewhere in this figure but for pairs of Sentinel-2 images from our test set with no plume embedded. This yields a false leak detection rate of 0.9%, which is in agreement with a pixel-wise false positive rate estimated below 0.03% (as shown in Fig. 2).

(iii) The red bin shows our deep learning model's detection rate over the regions surveyed in the Carbon Mapper catalog, but at

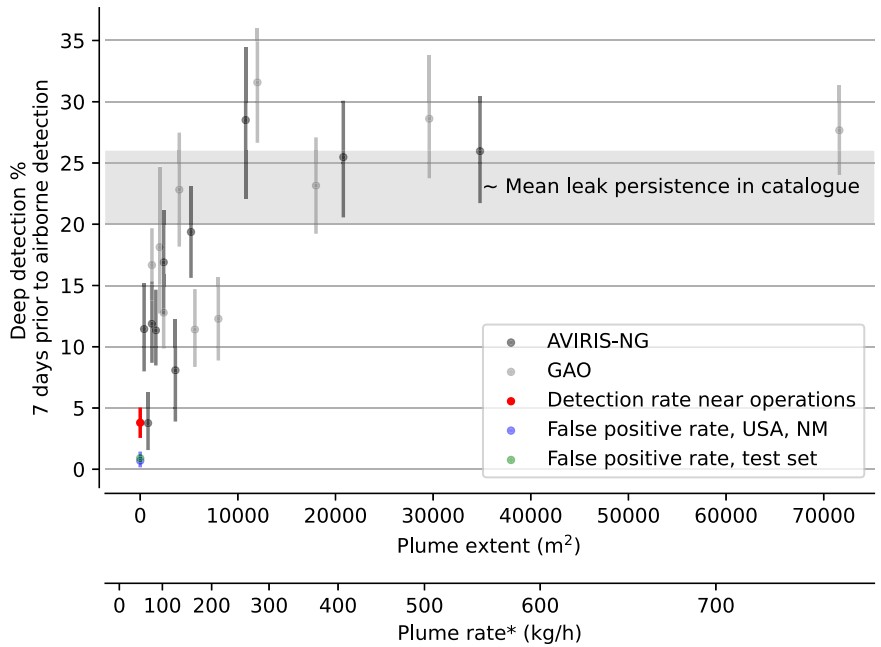

**Fig. 3 | Sentinel-2 detection of plumes cataloged by NASA's AVIRIS-NG and ASU's GAO airborne platforms.** In black and gray: fraction of cataloged plumes detected by our deep learning model in Sentinel-2 data. For each methane leak in the Carbon Mapper catalog, we retrieve $2.5 \times 2.5$ km$^2$ Sentinel-2 scenes centered on the known leak's location, that we feed to our trained model. We consider here that our model detects the corresponding leak if its output is above a threshold for two or more contiguous pixels within 500 m of the cataloged leak. Each black (resp. gray) dot and bar shows the mean and standard deviation of 332 (resp. 442) applications of our deep learning model in the 7 days preceding a cataloged detection using AVIRIS-NG (resp. GAO), for a total of 7724 applications to 2526 unique cataloged methane plumes. In blue (resp. green): exact same application of our model to data from the testing set with no embedded plume (resp. in Southern New Mexico outside of the Permian), illustrating the associated false positive detection rate. In red: application to the areas surveyed by the Carbon Mapper airborne campaigns, at random times and places. Data from Carbon Mapper's repository[8] of NASA's AVIRIS-NG and ASU's GAO detections in their 2020–2021 campaigns in California[41], Texas[36], Colorado, Arizona, Utah, Louisiana, and Pennsylvania. Our method detects most individual methane plumes in Sentinel-2 data down to 10 000 m$^2$ (corresponding on average to 200 to 300 kg/h leak rates), as our detection rate is close to the average persistence (20 to 26%) of the leaks[36] that are in the airborne catalog.

random locations and times (instead of centering the model's input on known leaks). This detection rate essentially shows the chances of detecting a different leak (and/or making a false detection) at random when assessing the detection of a particular leak of interest in an area containing methane sources.

Figure 3 shows that our model's performance is sensitive to the extent of the plume (as derived from the airborne detection) more so than the cataloged plume rate inversion (see Supplementary Fig. 8), with a clear breaking point at 10,000 square meters. The plume rate shown in the figure gives an estimate of an average leak rate for a given plume extent, and stems from a simple regression of leak rate versus plume extent in the Carbon Mapper catalog (see Supplementary Fig. 2). The cataloged leak location we use in Fig. 3 is the centroid of the cataloged plume mask.

Figure 3 shows that the Carbon Mapper cataloged plumes are reliably detected by our algorithm in Sentinel-2 data down to about 200 to 300 kg/h, as the fraction of cataloged leaks we detect is close to the average leak persistence in the catalog, which is the fraction of leaks that should still be there on average when observing them on a different day (the satellite observation being on a different day than the airplane observation). Our detection capabilities are close to the observation limit (the fraction of persistent leaks) down to 200 to 300 kg/h, after which our detections sharply drop off, with the smallest detected plumes being around 60 kg/h. Residual false detections are mainly due to clouds, rivers, and changes in soil moisture, and will be addressed in future work. Finetuning on specific regions of interest could also be performed in order to improve performance and limit false positives when using the model in practice. These results show that by leveraging a deep learning model trained on large amounts of Sentinel-2 data, observing methane in Sentinel-2 data is not restricted

to very large leaks and/or very bright surfaces[8], and that a trained model can detect plumes down to about an order of magnitude lower than the lower bound of previous Sentinel-2 methane detections solely based on band ratios[17,18].

Figure 4 shows examples of the application of our model to several methane plumes cataloged in the Carbon Mapper dataset, for various regions and leak rates. The plumes' leak rates and leak rate uncertainties are those from the catalog. The task of our deep learning model is only to detect the location of methane plumes in Sentinel-2 data, and the inversion for plume rate could be done as an additional step, using standard inversion methods such as the IME method[9] or by fitting a Gaussian plume[37]. The white crosses indicate the cataloged leak location, and our detections are in shades of orange. In the examples shown, inspection of optical imagery suggests that the emission sources correspond to oil and gas infrastructure (e,g,h,i), a pipeline (b), natural gas plants (c,f), an oil field (j), a coal mine (d), and a dairy farm (a).

Finally, a further test of our algorithm on data from a recent methane controlled release experiment[38] is shown in Figs. S3 and S4 of the Supplementary Materials. In this test, we show that our model can blindly detect the 4 controlled releases timed with Sentinel-2 passes, including a 1.1 ton/h leak that was missed by most groups that participated in the experiment. Our model makes no a priori on the location of the leak and has no knowledge of the local wind, and outputs little to no false positives off the plume.

Methane detection in global high resolution multi-spectral data (Sentinel-2, Landsat 8, etc.) has so far been limited to large emitters above 1 ton/h and has relied on thresholds on band ratios that generally require manual masking or manual verification[17,18]. Here we showed that deep learning models can provide an alternative that is

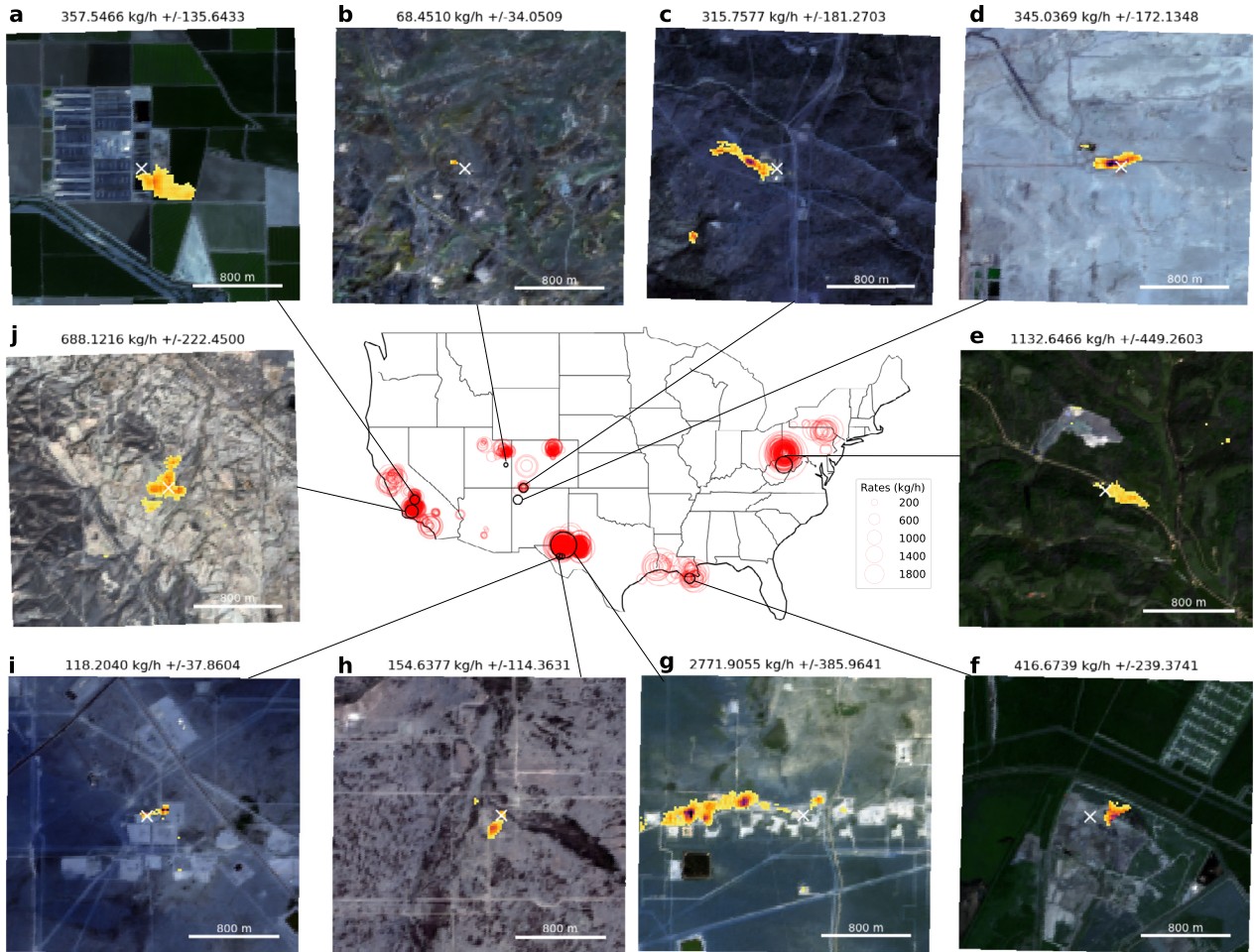

**Fig. 4 | Examples of Sentinel-2 deep learning detection of cataloged leaks.** All the 2526 leaks cataloged in the airborne Carbon Mapper dataset are shown in red on the map. The images surrounding the map show the application of our model to a few examples (black circles on the map), selected in various regions. In each of the images, the location of the cataloged plume is shown with a white cross in the center of the image, and the plumes detected by our model are shown in shades of orange, overlaid on the natural colors from bands B2, B3 and B4 from the same Sentinel-2 sample (the second acquisition fed to our model, see Fig. 1). The methane leaks can be attributed to oil and gas infrastructure (**e, g, h, i**), a pipeline (**b**), natural gas plants (**c, f**), an oil field (**j**), a coal mine (**d**) and a dairy farm (**a**). The leak rates and uncertainties shown here come from the rates in the catalog.

more robust to background noise and yields a drastically reduced false positive rate, opening the way towards the automation of methane detection at global scale.

Here we showed direct evidence on real Sentinel-2 data embedded with synthetic plumes that deep learning models are about an order of magnitude more robust to background noise (Fig. 2). In comparing the deep learning detections with airborne detections of real methane leaks in the U.S, we showed evidence, albeit indirect due to the time difference between satellite and airborne acquisitions, that this one order of magnitude improvement carries over to an operational setting (Figs. 3 and 4). Finally, on controlled releases of real methane plumes, we directly showed that deep learning can be used to automatically and blindly detect large methane emitters (Figs. S3 and S4).

## Discussion

Current global methane emissions monitoring systems rely on a layered and heterogenous approach that may start with global low resolution hyperspectral satellites that guide finer observations such as targeted satellites or airborne campaigns. A timely global monitoring system for methane, able to detect emissions down to a few hundreds of kilos per hour, would be a fundamental stepping stone towards an automated, complementary, and holistic monitoring system to building inventories of anthropogenic emissions at scale. In particular, such emissions account for the vast majority of U.S. methane point-sources in volume from methane-emitting areas analyzed in a recent survey of airborne campaigns in several States. By drastically lowering detection capabilities in multi-spectral data, our results suggest that public, general-purpose multi-spectral satellites can be turned into powerful methane monitoring tools capable of reaching detection performances approaching that of hyper-spectral constellations built specifically for methane detection, with the potential of generating global methane inventories at fine spatial and temporal scales.

Previous research has shown that the detection and remediation of a relatively small number of methane leaks can have a very large effect on reducing anthropogenic methane emissions[8]. However, finding these leaks amounts to finding a needle in a haystack, as the possible sources are innumerable and scattered around the globe, and except for the largest leaks, the methane plumes involved are only a few hundred meters in scale. This work represents a step towards the precise, systematic monitoring of methane emissions, anywhere on Earth, every few days. Moreover, the upcoming launch of Sentinel-2C and 2D in 2024 and 2025, respectively, and the inclusion of satellites from NASA's Landsat constellation in future work, should enable our method to reach a near daily global detection capability. Future

developments of our method will include further reducing the remaining false positives, notably by building ensembles of models and incorporating auxiliary data (e.g., water cover, wind, cloud masks, overall atmospheric methane concentration, etc.), and combining our methane detections with source rate quantification methods. In order to scale-up towards global detection of point sources of methane, we will implement our entire pre-processing and deep detection in the cloud. Last but not least, future work will also include collaborating to assess our method on methane controlled release experiments timed with Sentinel-2 and below 1100 kg/h.

We showed that our approach has the potential to provide global high frequency and high resolution detection of a large fraction of emitted methane from persistent point-sources, and we hope that its use will provide a building block towards the systematic quantification and accounting of methane, ultimately helping with the prioritization and validation of atmospheric methane mitigation.

## Methods

### Database of Sentinel-2 images
We download L1C Top-of-Atmosphere (TOA) reflectance data captured by ESA's Sentinel-2 constellation (Sentinel-2A and Sentinel-2B) through PEPS. We performed a visual inspection of the optical and landcover extent of the selected tiles to avoid, as much as possible, potential sources of point-source methane emissions. L1C TOA images are composed of 13 different spectral bands; of particular interest is band 12 in the SWIR, which is the only Sentinel-2 band that strongly overlaps with the methane absorption spectrum (along with band 11, but to a much lower extent, which we neglect). The spatial resolution of these images ranges from $10 \times 10$ to $60 \times 60$ m$^2$ per pixel according to the spectral band considered, with a $20 \times 20$ m$^2$ resolution in band 12. All spectral bands are re-sampled to the 20 m resolution of band 12, using nearest neighbor resampling. We keep the resolution of the SWIR bands unchanged.

We download time series of L1C images from various regions chosen to encompass a variety of climates (deserts, humid and vegetated areas), topographies, and land cover (water bodies, forests, etc.), avoiding as best as we can areas containing known or potential methane point sources (e.g., oil and gas fields or landfills). Because clouds are opaque to our signals of interest, we avoid Sentinel-2 tiles that contain more than 25% cloud cover (as reported in the images' metadata). This results in a total of about 900 pairs of $110 \times 110$ km$^2$ tiles. These tiles are then sub-divided into smaller windows of $2.5 \times 2.5$ km$^2$, and saved into HDF5 files for faster sampling during training.

### Generation of synthetic methane plumes
Deep learning models require a large number of examples to be trained in a robust manner, and physics-based plume simulations such as WRF-LES models[22] are slow to run and would not enable us to generate a sufficiently large amount of data. We opt instead for our own, simplified simulations, based upon Gaussian plumes[39]. The plumes are generated as follow:
- We randomly select an emission rate and wind velocity.
- We generate the associated Gaussian plume using the Gaussian plume model analytical solution[39].
- We add 2D colored noise to the generated plume, with the goal of mimicking atmospheric turbulences.

We generate about 20,000 of such plumes, and make sure to divide them into separate datasets for training, validation, and testing (one synthetic plume is never used both in training and testing). Note that information on emissions rates, etc. is not preserved, as the task of our deep learning model is only to detect Sentinel-2 pixels that contain methane.

### Embedding of synthetic plumes into Sentinel-2 images
The last step in building the training, validation, and test data consists in embedding those synthetic plumes into a portion of the processed Sentinel-2 L1C data. The plumes are embedded into the spectral band 12 of these images using the Beer-Lambert law.

Half of our training, validation, and testing data are augmented with synthetic plumes (positive examples), and half are kept without any modification (negative examples). The same Sentinel-2 image is never used in training and testing simultaneously (and furthermore are sampled from different regions of the globe). This results in a final training database of about 1,235,000 unique Sentinel-2 samples, a validation database of about 165,000 samples, and a testing database of about 245,000 samples.

### Model architecture and training
Our deep learning model has a U-net architecture, with its encoder portion replaced by a ViT, and its decoder left as a convolutional layers, with matching up-sampling using deconvolutional layers. The ViT portion of our model is the base variant introduced in the original paper[26], with a patch size of 16, which we train from scratch along with the decoder portion of the model.

We note that our architecture design stems from trial and error and outperformed more traditional purely convolution U-net architectures (see Supplementary Fig. S6). A more systematic exploration of model architectures will be the topic of future work.

Our deep learning model is trained on batches of 64 samples of pairs of $128 \times 128$ Sentinel-2 tiles, in which random Gaussian plumes are embedded. The model is trained for 10 epochs using the Adam variation of stochastic gradient descent[40], with a learning rate that starts at $10^{-3}$ and is progressively reduced by 0.1% when there is no improvement in validation performance after 10 batches. The model that has the best performance on the validation set is kept.

### Multi-band multi-pass methane detection method
MBMP results shown in Fig. 2 are based on the equation below, adapted from[17]:

$$\mathrm{MBMP}(t) = \frac{a_{12}^t B_{12}^t - a_{11}^t B_{11}^t}{a_{11}^t B_{11}^t} - \frac{a_{12}^{t-1} B_{12}^{t-1} - a_{11}^{t-1} B_{11}^{t-1}}{a_{11}^{t-1} B_{11}^{t-1}}, \qquad (1)$$

with $a_{12}^t, a_{11}^t, a_{12}^{t-1}, a_{11}^{t-1}$ normalization coefficients of the corresponding band and time. The methane detection derived from the MBMP method and compared against throughout the paper is then a threshold on the opposite of Eq. (1) (a decrease in B12 reflectance can indicate methane), with a threshold of 0.5, shown on the ROC curve, such that for high SNR the MBMP-based methane detector approaches a perfect F1 score (as seen on Fig. 2).

### Validation metrics on synthetic data
The classification error in Fig. 2 is assessed with the F1 score, the harmonic mean of precision and recall:

$$\mathrm{F1} = \frac{2}{\mathrm{precision}^{-1} + \mathrm{recall}^{-1}} = \frac{2\mathrm{TP}}{2\mathrm{TP} + \mathrm{FP} + \mathrm{FN}} \qquad (2)$$

with TP the true positives, FP the false positives and FN the false negatives.

The classifier built from our deep learning model is correspondingly when our model's output is above a certain threshold (0.2 in Fig. 2a, corresponding to the blue dot in Fig. 2b).

## Data availability
All the Sentinel-2 data used here is freely available from the European Space Agency on various repositories, such as PEPS used for this study

(https://peps.cnes.fr). The Carbon Mapper dataset is open-source and avalaible at: https://doi.org/10.5281/zenodo.7072824.

## Code availability

The code to reproduce the figures from the manuscript is available at: https://codeocean.com/capsule/6537045/tree/v1.

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

## Acknowledgements
This manuscript contains modified Copernicus Sentinel data (2020 to 2023). This work was initially funded by the U.S. Department of Energy under the SBIR grant DE-SC0022398. B.R.L. also acknowledges support from the JSPS Hakubi program. C.H. also acknowledges support from NASA's Entrepreneurs Challenge and the Entrepreneurial Research Fellowship administered by Activate Global, Inc. All opinions expressed in this paper are the authors' and do not necessarily reflect the policies and views of Activate Global, Inc.

## Author contributions
B.R.L. designed the method, trained the model and applied it to the Carbon Mapper catalog. C.H. designed the method and the model, and prepared the training data. Both authors analyzed the results and wrote the manuscript.

## Competing interests
The authors declare no competing interests.
