## [Peer Review File · Nature Communications]

Automatic Detection of Methane Emissions in Multispectral Satellite Imagery Using a Vision TransformerReviewer #1 (Remarks to the Author):

The authors present a novel vision transformer model to automatically detect methane plumes in Sentinel-2 satellite imagery. They report an order of magnitude improvement in detection limit relative to the current state of the art, down to 200 kg per hour. This is potentially a very important contribution, as the volume of methane-relevant satellite data has recently ballooned and now far surpasses the ability of human analysts to parse it. As exciting as the work is, however, I believe substantial revisions are needed before it can be accepted for publication. I have three main concerns:

First, I do not feel the results at present are sufficiently verified. The analysis of Figure 3 does not require overlapping passes between aircraft and satellite, which makes it of limited use for verification (more on this below). It is not clear to me whether the same is true for Figure 4. If the example plumes in Figure 4 can be directly compared with Carbon Mapper plumes on the same days, then the authors should include the side-by-side comparisons in the supplement. That would give some confidence in their detections. What I would really like to see, however, is a systematic comparison with the controlled releases performed by Sherwin et al. (2022). Several of those releases were timed with Sentinel-2 passes, and the emission rates ranged from ~1000-6000 kg per hour. Those plumes should be easily detectable by the authors' transformer model, and there were just a handful of them, so this is not asking a lot. A side-by-side comparison of the detected plumes with the Sherwin et al. plume images and ground-truth emission rates could provide strong support for the proposed model.

Second, Figure 3 raises several questions:

1. If the model is being applied to more than 2500 Carbon Mapper point sources, then why are there so few black points? Is some binning of sources being done?
2. How can there be 950 applications in three months prior to each Carbon Mapper detection when Sentinel-2 passes at most once every 2 days?
3. The transformations applied to the x-axis complicate interpretation of the results. If I understood correctly, the independent variable is not actually the catalogued Carbon Mapper plume rate but rather a prediction of the plume rate from the catalogued plume extent. But looking at supplemental Figure 2, this prediction is quite uncertain. The figure tries to say something about source strength, but it may say more about plume size, which is itself a function of source strength but also wind speed, topography, etc. I would suggest showing the comparison with the plume rates catalogued by Carbon Mapper somewhere in the manuscript.
4. Perhaps most importantly, the two caveats pointed out in the text (L. 183-192) are major ones. The Carbon Mapper surveys are predominantly in oil and gas fields, where sources are highly intermittent and clustered. Cusworth et al. (2021) reported a very wide range of persistence rates for Permian point sources, from ~0.1-0.9. The probability of the automatic detector marking "at least two adjacent pixels within 500 m of the leak" – i.e., within a 1-km disk – as positive for methane could be relatively high in the Permian basin, especially over a 3-month period. It is certainly not the same thing as the probability of the detector spotting the plume from a particular source. This concern could be addressed by applying the detector to random locations in the Permian – not merely to a location in the testing set, which is comprised of areas without known methane emitting infrastructure. How would that compare to the blue "false positive rate" point?

Third, the authors should attempt to explain why this new approach works so much better than previous ones – not just thresholded MBMP retrievals, but also U-Net detectors for hyperspectral instruments like PRISMA (Joyce et al., 2023) and GHGSat (Bruno et al., 2023), which one would expect to provide better methane detection than the spectrally much coarser Sentinel-2 but which often fail to detect larger plumes. Is self-attention the silver bullet here? Did the loss function or optimizer play a role? What is the input feature importance? I wonder whether the inclusion of reference imagery from a previous time $t-1$ might be a crucial element. More discussion of why the proposed approach is so successful would be appreciated.

Some more specific questions/comments:

1. Can the vision transformer approach be used to quantify methane plumes? I was left wondering what one would do to quantify emissions after using the proposed model to detect the plumes.

Perhaps this is future work, but it could use some discussion.

2. L. 68 and L. 230: The claim about 200 kg/h and “the vast majority” of methane from point sources seems to refer to Figure 7 from Jacob et al. (2022), but that was for US aerial surveys, mostly of oil and gas basins. One of the surveys in that plot, for California, shows >200 kg/h accounting for less than half of the point source budget – and we don’t know much about the source distribution in other countries/industries. Some caution is needed here.

3. L. 97 and L. 283: Please say more about this auto-correlated noise for reproducibility.

4. L. 97-98: “further emulate atmospheric turbulence” – This should be rephrased for clarity. The Gaussian plume model doesn’t emulate turbulence at all, so adding auto-correlated noise does not “further” do that. Same issue on L. 283 with “further mimicking”.

5. Fig. 2 caption: “median of 40 bins” – What are these bins?

6. L. 274: What is this “data-friendly” format?

Typos

- L. 28 in combination to -> in combination with
- L. 67 models -> model
- Fig. 1 caption: as two different times -> at two different times
- L. 194 leaks -> leak
- L. 196 leat -> least
- Fig. 4 caption: showed -> shown

References

Cusworth, D. H., Duren, R. M., Thorpe, A. K., Olson-Duvall, W., Heckler, J., Chapman, J. W., Eastwood, M. L., Helmlinger, M. C., Green, R. O., Asner, G. P., Dennison, P. E., and Miller, C. E.: Intermittency of large methane emitters in the Permian Basin, *Environ. Sci. Technol. Lett.*, 8, 567–573, <https://doi.org/10.1021/acs.estlett.1c00173>, 2021.

Jacob, D. J., Varon, D. J., Cusworth, D. H., Dennison, P. E., Frankenberg, C., Gautam, R., Guanter, L., Kelley, J., McKeever, J., Ott, L. E., Poulter, B., Qu, Z., Thorpe, A. K., Worden, J. R., and Duren, R. M.: Quantifying methane emissions from the global scale down to point sources using satellite observations of atmospheric methane, *Atmos. Chem. Phys.*, 22, 9617–9646, <https://doi.org/10.5194/acp-22-9617-2022>, 2022.

Sherwin, E. D., Rutherford, J. S., Chen, Y., Aminfard, S., Kort, E. A., Jackson, R. B., and Brandt, A. R.: Single-blind validation of space-based point-source detection and quantification of onshore methane emissions, *Sci. Rep.-UK*, 13, 3836, <https://doi.org/10.1038/s41598-023-30761-2>, 2023.

Reviewer #2 (Remarks to the Author):

The authors present a new deep learning approach to detect methane plumes in Sentinel-2 multi-spectral satellite imagery. They construct a large training dataset by downloading Sentinel-2 data and synthesizing methane plumes to serve as training labels, then evaluate the approach on a set of real plume data. They find the model can detect many of the real plumes without identifying a substantial amount of false positives, suggesting the potential for the approach to serve as a cheap, efficient, and accurate alternative to other methane plume detection approaches.

While I deeply appreciate the mission of the work and believe that the approach has potential, I see several major issues with both the methodology and conclusions made from the results, so I cannot recommend the paper for publication. I’ve provided my major and minor comments about the work below.

Major Comments:

1. There are several claims made in the paper which are poorly supported by the results presented.

1(a). Importantly, I am not convinced from the results that the major conclusion of the paper (the only result highlighted in the Abstract, lines 20-23), that the model can “reliably detect” the point

sources down to 200kg CH₄ h⁻¹, is true.

1(a)(i). Figure 3 shows that the model detected ~80% of the expected visible plumes between 200 and 350 kg/h then about 90-95% between 400 and 1000. Missing 20% of plumes seems substantial. Without a baseline or reference method to compare against, claiming this is reliable detection is subjective. Furthermore, determining these aggregate numbers of how many plumes were detected above some plume rate, which are key to the main conclusions of the paper, was not easy to do from the figure. Those should be presented numerically or at least shown more clearly in a revised or additional figure.

1(a)(ii). The false positive rate was approximated and therefore it is not clear how reliably the model will detect plumes in practice. Furthermore, even if the estimated false positive rate of 3-4% is true, that corresponds to at least 2,400 false positives 80,000 images. The authors state a few times that future work will focus on reducing these, but the limitation should be acknowledged more explicitly, and the major conclusion should be tempered accordingly.

1(b). Lines 69-70 state their approach unlocks fully automated monitoring of emissions, but the approach produces a nontrivial amount of false negative and false positives detections which may prohibit practical use. This claim needs to be changed or better supported by the analysis in the paper.

1(c). The authors make the claim on lines 82-84 that the database contains areas that do not encompass known potential methane sources, but they do not thoroughly describe how they ensured this while sampling, they do not supply an exhaustive list of potential methane sources, and they state they were only measured far from oil and gas activity on line 160, which I believe puts into question whether the claim on lines 82-84 is true. The sampling process should be described in more detail, and if the authors believe there could be known potential sources in the negatives, this claim should be modified and conclusions made from results should be adjusted accordingly if needed.

2. Several methodological decisions were made without sufficient motivation and explanation, some of which may affect the results and conclusions made from the paper.

2(a). Why use a single pair of consecutive times? Why only predict a single plume image? Does the plume correspond to one of the times, the difference, or something else?

2(b). Why are all input bands sampled to 20m? How is the downsampling for 10m bands and upsampling for 60m bands done?

2(c). What operation is used to perform the upsampling in the UNet architecture?

2(d). Why did the authors design and use the proposed deep learning architecture rather than other well-established architectures? ViTs are known to struggle with dense prediction tasks, so comparing to well-established CNN-based segmentation architectures and ViT-based segmentation architectures that have been modified for dense prediction would much better support the use of this architecture, should it outperform those approaches. Furthermore, several implementation details were left out, like the optimization procedure (optimizer, learning rate, batch size, etc.) for example.

2(e). Why split datasets by region if the data is synthetically generated? Wouldn't one want to include all regions in the training set to improve the generalization, rather than excluding some regions?

2(f). Regarding the argument made on lines 178 - 181, even though training on synthetic data was not centered around the plumes, doing so when testing the model is biased. If the goal is to understand how well the model works for detecting real methane plumes in practice, the locations of the methane plumes and often even the potential sources will not be known apriori and therefore the images will not be centered on the plumes. The evaluation needs to be redone without centering it in order to identify whether the model is robust to perturbations in the locations of the real methane plumes.

3. No limitations are stated even though there are several, including but likely not limited to (1) the use of randomly sampled negatives which may have methane emissions, which is likely reducing the sensitivity and leads to a rough approximation the false positive rate, (2) the approximated false positive rate is nontrivial, (3) the uncertainty of real-world detection accuracy arising from intermittency of plumes.

Minor Comments:

Overall

1. The syntax used throughout the paper is good. However, paragraph organization and structuring could use work. For example, why are lines 110-118 three separate paragraphs? Why are lines 218-224 separate paragraphs?
2. The authors are not consistent with spelling "multi-spectral", sometimes excluding the hyphen. It would be good to make this consistent and look for other potential inconsistencies throughout the paper.
3. The Figures (specifically 1 and 3) could be substantially improved to improve readability.
3 (a). Figure 1 could more clearly show how the images are being divided into patches, which is an important design decision. It could also be more aesthetically designed overall, including using visual representations of the operations that are more standard (and in my opinion, more clear).
3 (b). In addition to the changes I suggested in the major comments, Figure 3 could include horizontal gridlines to improve readability.
4. The Data availability and code availability suggest neither the curated dataset nor model training code will be made publicly available. However, the dataset and code would be fantastic resources for other researchers to be able to use and, in my opinion, releasing them would lead to the work making a much larger impact.

Abstract

1. The Abstract could cite more specific numbers, e.g. the capture rate.

Introduction

1. The background and motivation for the work is solid. However, the claims made in the last paragraph are exaggerated. Lines 67-70 again claim that the models robustly detect emissions down to 200kg/h, and that it unlocks fully automated monitoring, which I don't think the current results sufficiently support as I've explained above.

Results

1. It'd be helpful to plot the locations of the training / validation / test tiles on a map, at least as a supplementary figure, so that readers can gain a better understanding of the geographic distribution of the splits.
2. References to Fig. 2B and 2C are swapped in the text from lines 139-157 (or the figure/caption labels need to be swapped)
3. The model is not a transformer. It is a semantic segmentation model with a vision transformer encoder. I'd suggest modifying language around this throughout the paper (e.g. lines 102, 110), potentially including the title.
4. Line 114 and the caption state that the model is an auto-encoder but it is not. An auto-encoder attempts to reconstruct the input. This is an encoder-decoder segmentation architecture.
5. Lines 158-160 claim the FPR is reliably evaluated, but I think the authors are trying to say that their estimated FPR is likely an upper bound of the true FPR, given some of the detected plumes may actually be real plumes, suggesting it is not reliably evaluated. Additionally, for the exact reason the FPR is likely overestimated, the FNR may be underestimated, which could be inflating the AUC.
6. Why switch between kg/h and tons/h on line 167? Better to stay consistent with one (or report both).

Discussion

1. I think the discussion does a better job of making more supportable conclusions than other parts of the paper. However, the claim on lines 233-236 again seems too strong as the authors do not present results which sufficiently compare to these constellations. I'd suggest softening this claim or including these results.

Methods

1. How were the Sentinel-2 images downloaded?
2. How was cloud cover determined?

We are very grateful for the time and effort spent by the reviewers on our paper, and for their comments that helped us to drastically improve the manuscript. Notably, by following the input of the reviewers, we have further trained our model and applied it more conservatively such that our false positive rate has been reduced.

The supplementary materials have been expanded to assess the behavior of our model in a variety of circumstances. Last but not least, following the advice from reviewer 1, we have applied our model to a recent controlled release experiment organized by Stanford.

In the following, the reviewers' comments are in black and our answers are in light blue, and the edited text in the manuscript and its supplementary is in dark blue.

Reviewer #1 (Remarks to the Author):

The authors present a novel vision transformer model to automatically detect methane plumes in Sentinel-2 satellite imagery. They report an order of magnitude improvement in detection limit relative to the current state of the art, down to 200 kg per hour. This is potentially a very important contribution, as the volume of methane-relevant satellite data has recently ballooned and now far surpasses the ability of human analysts to parse it.

We thank the reviewer for these kind and encouraging words!

As exciting as the work is, however, I believe substantial revisions are needed before it can be accepted for publication. I have three main concerns:

We tried to answer these main concerns as well as possible; explanations and additional information, as well as individual changes to the manuscript and supplementary materials, are described in what follows.

First, I do not feel the results at present are sufficiently verified. The analysis of Figure 3 does not require overlapping passes between aircraft and satellite, which makes it of limited use for verification (more on this below). It is not clear to me whether the same is true for Figure 4. If the example plumes in Figure 4 can be directly compared with Carbon Mapper plumes on the same days, then the authors should include the side-by-side comparisons in the supplement. That would give some confidence in their detections. What I would really like to see, however, is a systematic comparison with the controlled releases performed by Sherwin et al. (2022). Several of those releases were timed with Sentinel-2 passes, and the emission rates ranged from ~1000-6000 kg per hour. Those plumes should be easily detectable by the authors' transformer model, and there were just a handful of them, so this is not asking a lot. A side-by-side comparison of the detected plumes with the Sherwin et al. plume images and ground-truth emission rates could provide strong support for the proposed model.

Thank you for your suggestion. Indeed, because the satellite passes and the Carbon Mapper airborne campaigns are not synchronized, it is impossible for us to directly compare the plumes detected by our algorithm to the plume masks reported by Carbon Mapper (this is true for both Figures 3 and 4). We have changed the text of the main manuscript to make it clearer that our argument is statistical, and we have toned down the claim that our model is able to detect every leak above 200 to 300 kg/h. Statistically, if our model is able to detect the Carbon Mapper plumes, its detection rate over the catalogue should be around the average persistence of the plumes (defined here and in the literature as the probability of a leak to be active during a random measurement). Because our detection rate has an asymptote towards this mean persistence for leaks above 200 to 300 kg/h,

while our detection rate sharply drops below that, we argue that our model is able to detect methane leaks above 200 to 300 kg/h.

We also modified Figure 3 to only consider Sentinel 2 images within 7 days of the airborne plumes' detections, to strengthen the comparison.

We have added the following summary in the main text to try to make it clearer:

(1.283) “Here we showed direct evidence on real Sentinel 2 data embedded with synthetic plumes that deep learning models are about an order of magnitude more robust to background noise (Fig. 2). In comparing the deep learning detections with airborne detections of real methane leaks in the U.S, we showed evidence, albeit indirect due to the time difference between satellite and airborne acquisitions, that this one order of magnitude improvement carries over to an operational setting (Figs. 3 and 4). Finally, on controlled releases of real methane plumes, we directly showed that deep learning can be used to automatically and blindly detect large methane emitters (Figs. S3 and S4).”

We added two new figures in the Supplementary Materials (Figures S3 and S4) analyzing the application of our model to the latest controlled methane releases performed by Sherwin *et al.* [1]. Figure S3 shows that our model is able to successively detect all four controlled releases timed with Sentinel-2 overpasses, and compares our detection with the masked detections made by the groups that participated in the blind test. Interestingly, while most groups missed the smallest plume (1.1 ton/h - the smallest leak ever detected in Sentinel-2 according to the literature), our algorithm is able to successfully detect it. Note that here again, there is no proper ground truth in terms of the plumes' masks, but the overall shape and direction of our retrieved plumes are very similar to those computed by the participating groups, and are coherent with the release location and the local wind direction (information that are not provided to our algorithm).

Figure S4 compares our detections with the unmasked methane retrieval fields computed by the groups participating in the blind test, and illustrates the amount of false positives typically present in state-of-the-art analyses. The detection of methane plumes in these methane retrieval fields typically involves a human operator, while our approach detects plumes automatically. We note also that for these masked detections, all groups used the information of the location of the release as well as the wind direction, *while our model detected the plumes completely blindly.*

Although our Carbon Mapper analysis cannot definitely prove automatic detection capabilities down to 200 kg/h (and only provides a statistical argument), we argue that this further analysis of the Sherwin *et al.* dataset proves that our algorithm can automatically and blindly detect leaks down to 1100 kg/h. We are very thankful to reviewer 1 for suggesting this additional analysis which greatly strengthens our manuscript, and we have changed the main text to argue for this distinction between a proven automatic detection capability down to 1100 kg/h and a likely capability down to 200 kg/h, which will require further testing against future controlled releases.

Second, Figure 3 raises several questions:

1. If the model is being applied to more than 2500 Carbon Mapper point sources, then why are there so few black points? Is some binning of sources being done?

Thank you for raising this question. Indeed, binning is performed – each of the bins corresponds to the application of our model to a number of pairs of Sentinel-2 images. In the new version of Figure 3, the second image of the pair is taken in the seven days before the leak, and the first image in the 3 months prior to the leak. To further analyze the behavior of our approach, we have separated campaigns conducted with two distinct instruments (AVIRIS-NG and GAO), and we find that our results are consistent for both instrument types. Each bin for AVIRIS-NG includes 332 applications

to pairs of Sentinel 2 images, and each bin for GAO 442. Each point shows the mean and the 95% confidence interval of the bin.

2. How can there be 950 applications in three months prior to each Carbon Mapper detection when Sentinel-2 passes at most once every 2 days?

This is because the model was applied on pairs of Sentinel-2 images, wherein each individual image is captured on a different day, with the reference date in the 3 months prior to the airborne detection, and the second detection date within 7 days of the airborne detection. Combining two dates enables to generate a much larger number of data points and to tighten confidence intervals, explaining the 7000 applications of our model described in the former Figure's caption.

In the previous version of the manuscript, both dates were within 3 months before the leak, but we have tightened this to 7 days so that the leak is more likely to be there.

We have added the following explanations to try to make it clearer:

(1.199) "This pair of Sentinel 2 images consists in a reference date in the 3 months prior to the leak, and a detection date in the 7 days prior to the leak. We further restrict our analysis to cloudless days (less than 0.5% cloud cover), which yields a total of 7724 possible pairs for the 2526 leaks of the catalogue."

(1.218) "The Carbon Mapper catalogue regroups data from campaigns performed with two instruments (AVIRIS-NG and GAO). Our detection results are summarized separately for the AVIRIS-NG (black) and GAO (grey) campaigns, in 10 bins, with each bin showing the average detection rate of our deep learning model, the average leak size and rate in the bin, for 332 and 442 Sentinel 2 pairs per bin, respectively"

3. The transformations applied to the x-axis complicate interpretation of the results. If I understood correctly, the independent variable is not actually the catalogued Carbon Mapper plume rate but rather a prediction of the plume rate from the catalogued plume extent. But looking at supplemental Figure 2, this prediction is quite uncertain.

The figure tries to say something about source strength, but it may say more about plume size, which is itself a function of source strength but also wind speed, topography, etc. I would suggest showing the comparison with the plume rates catalogued by Carbon Mapper somewhere in the manuscript.

Thank you for your comment. Yes, this is exactly right, we find empirically that our model's performance is mostly sensitive to the plume's extent, rather than the leak rate. There is indeed a rather uncertain relation between plume rate and plume extent, that also depends on wind and topography, and the fact that catalogued plume rate inversions themselves come with large uncertainties.

To make it clearer, we have changed the axes of Figure 3 to show the plume extent as the main driver for our model's performance; we show our rescaled leak rate as a guide, and indicate so in the text:

(1.238) "Fig. 3 shows that our model's performance is sensitive to the extent of the plume (as derived from the airborne detection) more so than the catalogued plume rate inversion (see Supplementary Fig. 8), with a clear breaking point at 10,000 square meters. The plume rate shown in the figure gives an estimate of the corresponding leak rate, and stems from a simple regression of leak rate versus plume extent in the Carbon Mapper catalog (see Supplementary Figure 2)."

We also added a new figure in Supplementary Material (Fig. S8) illustrating the relationship between our model's results and the plume rates reported by Carbon Mapper, with no rescaling performed, along with the following explanation:

(Suppl. Fig. 8) "Fig. 8 shows the fraction of catalogued Carbon Mapper leaks that were detected by our deep learning model, as a function of leak rate. Compared with Fig. 3 of the main text, we can

see that the main factor explaining detection is the plume's extent, rather than the plume's leak rate. However, plume extent and plume rate are closely correlated, with variations mostly due to wind conditions, and the plume's extent can be rescaled as an expected plume rate (using the fit from Fig. 1 of the supplementary), as shown in the secondary axis of Fig. 3 of the main text.”

4. Perhaps most importantly, the two caveats pointed out in the text (L. 183-192) are major ones. The Carbon Mapper surveys are predominantly in oil and gas fields, where sources are highly intermittent and clustered. Cusworth et al. (2021) reported a very wide range of persistence rates for Permian point sources, from ~0.1-0.9. The probability of the automatic detector marking “at least two adjacent pixels within 500 m of the leak” – i.e., within a 1-km disk – as positive for methane could be relatively high in the Permian basin, especially over a 3-month period. It is certainly not the same thing as the probability of the detector spotting the plume from a particular source. This concern could be addressed by applying the detector to random locations in the Permian – not merely to a location in the testing set, which is comprised of areas without known methane emitting infrastructure. How would that compare to the blue “false positive rate” point?

We have added such a false positive rate to Fig. 3 by applying the detector to random locations in the Permian and the other locations of the catalogue, which summarizes the odds of a making a random detection, instead of making a detection at the time and place of a known leak.

We note that when our neural network is fed a pair of Sentinel 2 images, it is only tasked with finding methane in the second image, with the first image being only used as a reference. Therefore, the odds of detecting a leak other than the one being assessed against should be independent from the detection date and the time difference between the reference date and the detection date. We have also added another test away from the Permian Basin, but in a similar environment, near Las Cruces in southern New Mexico. This gives us a false positive rate in similar environments as that of the Permian, that can be directly compared with the detection rate of other leaks at random in the Permian and other areas surveyed in the Carbon Mapper catalogue.

We have added the following text to describe these tests:

(l.224) “Fig. 3 also summarizes our detection rate when applying the exact same methodology in the absence of known leaks, for three different tests. i) The blue bin shows our average detection rate when applying our model to pairs of Sentinel 2 images over southern New Mexico (but away from the Permian Basin), and provides an estimated false positive rate of 0.7% in conditions similar to that of the Permian Basin. ii) The green bin shows our false detection rate using the exact same methodology as elsewhere in this figure, but for pairs of Sentinel 2 images from our test set with no plume embedded. This yields a false positive rate of 0.9%, which is in agreement with a pixel-wise false positive rate estimated below 0.03% (as shown in Fig. 2).

iii) The red bin shows our deep learning model's detection rate over the regions surveyed in the Carbon Mapper catalogue, but at random locations and times (instead of centering the model's input on known leaks). This detection rate essentially shows the chances of detecting a different leak (and/or making a false detection) at random when assessing the detection of a particular leak of interest in an area containing methane sources”.

Third, the authors should attempt to explain why this new approach works so much better than previous ones – not just thresholded MBMP retrievals, but also U-Net detectors for hyperspectral instruments like PRISMA (Joyce et al., 2023) and GHGSat (Bruno et al., 2023), which one would expect to provide better methane detection than the spectrally much coarser Sentinel-2 but which often fail to detect larger plumes. Is self-attention the silver bullet here? Did the loss function or optimizer play a role? What is the input feature importance? I wonder whether the inclusion of reference imagery from a previous time t-1 might be a crucial element. More discussion of why the proposed approach is so successful would be appreciated.

Thank you for raising this question, that we indeed did not address enough in the manuscript. We believe that the performance of our algorithm is due in priority to two factors:

- First, the reliance on simple Gaussian models to generate synthetic methane plumes to train the model. To our knowledge, all of the deep learning approaches currently published with the goal of recovering methane signatures in spectral data rely either on real plumes, or on WRF-LES simulation schemes to train the models. This is a major issue, because the resulting training datasets are way too small to train a robust model, typically on the order of a few thousand of examples. There are currently not enough real methane plumes in existing databases (and using real plumes introduces the issue of segmenting them first). The WRF-LES simulations are too slow to generate a large database for training – the same simulated plume can be captured at various times to improve the number of examples, but this still generates issues in terms of the diversity of the signals. In contrast, the simple Gaussian models take a fraction of a second to run, which enables us to generate large numbers of diverse signals, thereby leading to a larger and varied training dataset that enables to train a more robust model. In particular, Joyce et al., 2023 use a training/testing dataset of 9700 images, compared to our 550,000 image pairs in the original submission and more than 1,500,000 image pairs in the revised submission. We argue that less than a few hundred thousand samples is not enough data to properly train a deep learning model. Similarly, Bruno et al. 2023 train their deep learning model on a database of 6870 samples.
- Another critical explanation for the success of our method is indeed the use of two dates for the input of our model, with a reference data and a detection date, such that the deep learning model is free to learn an optimal methane retrieval method. The conceptual idea is very similar to the MBMP method. With respect to methane, there are many sources of noise in Sentinel-2 or other spectral data that can be drastically lowered when comparing the same location at two different times. In particular, surfaces that absorb in SWIR bands will have similar reflectance at both reference and detection dates (which explains why the comparison of band ratios at different times in MBMP enables to remove a large portion of the noise). In contrast, the presence of methane at the detection date only will reduce reflectance in band 12 but not in band 11, and only at the second date. Feeding two images to the algorithm enables it to distinguish persistent absorption in SWIR at a given location (most likely noise w.r.t methane signals) from non-persistent absorption in band 12 only (possible methane absorption signals). We initially built models using a single time, but these performed a lot more poorly.

Relying on transformer models instead of e.g. CNNs does improve performance, but this improvement is limited compared with the impact of the two points listed above.

We have added the following text in the discussion part of the paper to try to explain the performance of our approach:

(1.142) “We explain the strong performance of our model by three main factors: i) relying on Gaussian plumes randomly embedded in real Sentinel 2 data (instead of real plumes or computer-intensive WRF-LES simulations), enables us to generate a training dataset that is orders of magnitude larger than the ones typically used in previous attempts at developing deep learning models for methane detection, thereby enabling to fully train large deep learning models. ii) The use of two time-steps as input (conceptually similar to the MBMP approach) enables the model to use the first image as a reference image, to which the second image is compared in order to identify transient signals in methane absorbing band 12, while false positives in band 12 can be discriminated using the other bands and their evolution over the two time-steps. This comparison is crucial to distinguish signal from noise and correctly detect methane plumes, in particular the smaller ones. And iii) the use of transformers instead of convolutional neural networks (CNNs) enables our model to capture the long-range nature of a plume.”

Some more specific questions/comments:

1. Can the vision transformer approach be used to quantify methane plumes? I was left wondering what one would do to quantify emissions after using the proposed model to detect the plumes. Perhaps this is future work, but it could use some discussion.

Yes, this is possible, and some members in our team are currently working on this (but at this point we haven't finished to develop it). This has also been done in a couple of papers in the literature. Another approach is to use a plume mask, built on top of our model's output, in combination with standard quantification methods such as IME. We are currently favoring the second approach, as using deep learning for methane mask identification followed by a physical method for rate inversion somewhat avoids the pitfalls of the black-box nature of deep learning models. We have added the following text to discuss this in the paper:

(1.262) "The task of our deep learning model is only to detect the location of methane plumes in Sentinel 2 data, and the inversion for plume rate could be done as an additional step, using standard inversion methods such as the IME method or by fitting a Gaussian plume."

2. L. 68 and L. 230: The claim about 200 kg/h and "the vast majority" of methane from point sources seems to refer to Figure 7 from Jacob et al. (2022), but that was for US aerial surveys, mostly of oil and gas basins. One of the surveys in that plot, for California, shows >200 kg/h accounting for less than half of the point source budget – and we don't know much about the source distribution in other countries/industries. Some caution is needed here.

Thank you for raising this issue. We indeed used this number because it corresponds to the median of the various campaigns analyzed in Jacob et al. (2022), but it is true that emission distribution can vary drastically from one region of interest to the other, and our current description may be confusing in that respect. In particular, for one airborne campaign analyzed in California, the fraction reported in Jacob et al. (2022) would only be 30% (this number is still higher than 70% for the other Californian campaigns). We have also recently learned from presentations at AGU that leak size distribution in Alberta, Canada, is on average much lower than in the U.S., and the aforementioned statement would not apply to this region.

To clarify our results, we have modified both paragraphs in the main text:

(1.69) "Our results suggest that our model detects most methane emissions down to plumes of 0.01 km², corresponding to methane leak rates of 200 to 300kg/h (with variations depending on wind conditions). Leaks of this size account for the vast majority of the estimated methane budget coming from point sources for most airborne campaigns in California, Colorado, and the Permian Basin analyzed in a recent survey".

(1.297) "In particular, such emissions account for the vast majority of U.S. methane point-sources in volume from methane-emitting areas analyzed in a recent survey of airborne campaigns in several States."

3. L. 97 and L. 283: Please say more about this auto-correlated noise for reproducibility.

We add colored noise to the plumes in order to mimic turbulences. We have modified the synthetic plumes' description in the Methods section:

(1.354) "We add 2D colored noise to the generated plume, with the goal of mimicking atmospheric turbulences."

4. L. 97-98: "further emulate atmospheric turbulence" – This should be rephrased for clarity. The

Gaussian plume model doesn't emulate turbulence at all, so adding auto-correlated noise does not "further" do that. Same issue on L. 283 with "further mimicking".

Thank you, we have modified the text for greater clarity.

5. Fig. 2 caption: "median of 40 bins" – What are these bins?

In order to produce Fig. 2, 40 bins of equal number of samples along the SNR axis are built and the median F1 score in each bin is shown as the thick lines. We have added the following parenthesis to clarify this:

(Caption of Figure 2) "(of equal number of samples along the SNR axis)"

6. L. 274: What is this "data-friendly" format?

Data is saved in the hdf5 format. We have changed the text to be more specific, and removed the subjective 'data-friendly':

(1.345) "These tiles are then sub-divided into smaller windows of 2.5 x 2.5 km², and saved into HDF5 files for faster sampling during training."

Typos

- L. 28 in combination to -> in combination with
- L. 67 models -> model
- Fig. 1 caption: as two different times -> at two different times
- L. 194 leaks -> leak
- L. 196 leat -> least
- Fig. 4 caption: showed -> show

Thank you for catching these typos! We modified the text.

Reviewer #2 (Remarks to the Author):

The authors present a new deep learning approach to detect methane plumes in Sentinel-2 multi-spectral satellite imagery. They construct a large training dataset by downloading Sentinel-2 data and synthesizing methane plumes to serve as training labels, then evaluate the approach on a set of real plume data. They find the model can detect many of the real plumes without identifying a substantial amount of false positives, suggesting the potential for the approach to serve as a cheap, efficient, and accurate alternative to other methane plume detection approaches.

While I deeply appreciate the mission of the work and believe that the approach has potential, I see several major issues with both the methodology and conclusions made from the results, so I cannot recommend the paper for publication. I've provided my major and minor comments about the work below.

Thank you very much for the time spent on reviewing our manuscript and for your comments and suggestions, which helped us to markedly improve our results. We agree that the mission of automatizing methane detection is critical, and we have done a lot of further work since the initial submission to try to address the issues raised by Reviewer 2. In particular, we have retrained the model on 3 times more data, including on more challenging data (more vegetated areas, and cloudier data), which greatly improved the performance of our model by dramatically reducing its false positive rate.

We have also toned down our claims and conclusion, as indeed the comparison with the leaks catalogued by Carbon Mapper only gives us an indirect indication of our detection performance (because the aerial and satellite passes are not at the same time). We have also applied our method to 4 controlled methane releases that were done at the same time as the Sentinel 2 satellite passes, that gives a solid validation that our approach does automatize methane detection in the 1+ ton/h range.

Our point-by-point response follows.

Major Comments:

1. There are several claims made in the paper which are poorly supported by the results presented. 1(a). Importantly, I am not convinced from the results that the major conclusion of the paper (the only result highlighted in the Abstract, lines 20-23), that the model can “reliably detect” the point sources down to 200kg CH₄ h⁻¹, is true.

1(a)(i). Figure 3 shows that the model detected ~80% of the expected visible plumes between 200 and 350 kg/h then about 90-95% between 400 and 1000. Missing 20% of plumes seems substantial. Without a baseline or reference method to compare against, claiming this is reliable detection is subjective. Furthermore, determining these aggregate numbers of how many plumes were detected above some plume rate, which are key to the main conclusions of the paper, was not easy to do from the figure. Those should be presented numerically or at least shown more clearly in a revised or additional figure.

Thank you very much for your comment and suggestions. We did our best to modify the main claims of the paper and align them with our results. As mentioned above, we also spent a lot of time building a larger dataset and further training the model, and we were able to lower our false positive rate on the test set from 0.5% to 0.03% pixel-wise. This leads to much lower false positives as reported in Figure 3.

Methane detection in Sentinel 2 data is currently limited to 1 ton/h+ leaks (and often struggle to detect plumes below 8-10 tons/h, including in critical areas such as the Permian basin [3,4]). Current approaches consist in manual identification of plumes in very noisy methane retrieval fields such as the MBMP method, which we compare our method against (there is currently no method that automatically identifies methane plumes in Sentinel 2). Importantly in the context of this review, there are also no previous examples of methane leak detections in Sentinel 2 data below 1ton/hour, which makes the validation of our approach on small leaks particularly challenging.

To answer the comment more specifically, we only have a rough idea of the persistence of leaks in the catalog (the 20-26% estimate indicated in the figure), and it is not clear how persistence varies in time and with leak rate. The persistence of leaks and their variation in time is an active area of research (see for example Cusworth et al., Environ. Sci. Technol. Lett. 2021, 8, 7, [2]), and beyond the scope of our paper.

In order to try and make the comparison more robust, we have modified our figure to only consider satellite detection within 7 days of the airborne detection (compared with 3 months in the previous version of the manuscript). This enables us to detect a larger fraction of the leaks while at the same time having a more conservative detection threshold. However, because our model is still applied to data up to 7 days apart from the airborne detection, we cannot guarantee that any particular leak is still there. The goal of this exercise, and that of Fig. 3, is to show a clear statistical relationship between our satellite detections and the airborne detections for leaks below 1 ton per

hour. We have toned down our wording however, as indeed our comparison with airborne detection only gives us an indirect proof and is not enough to claim “reliable detection”.

Furthermore, we added an analysis of a controlled release methane experiment in the Supplementary Materials. In this recent experiment conducted by Serwin et al., four 1ton/h+ methane releases were timed with Sentinel 2 passes. One of these plumes (the 1.1 ton/h release) corresponds to the smallest plume ever detected so far in Sentinel 2 data in the literature.

Figures S3 and S4 shows the application of our model to these controlled methane releases, with our model having once again no information on the location of the plume nor the local wind (which was given to the participants of the test). Figure S4 also shows methane retrieval fields computed by the groups participating in the blind test, and illustrate the amounts of false positives that are present in state-of-the-art analyzes. Masked detections from these retrieval fields typically involve analysis by a human operator.

Throughout the manuscript, we have toned down our claim that our model can “reliably” detect methane, replacing it with the claim that our model is “significant step towards automatically” detecting most methane point sources.

1(a)(ii). The false positive rate was approximated and therefore it is not clear how reliably the model will detect plumes in practice. Furthermore, even if the estimated false positive rate of 3-4% is true, that corresponds to at least 2,400 false positives 80,000 images. The authors state a few times that future work will focus on reducing these, but the limitation should be acknowledged more explicitly, and the major conclusion should be tempered accordingly.

Following this comment, we have been working hard for the last 3 months at reducing the false positive rate of our approach. Notably, we have trained the model on 3 times more samples (1.6 million samples compared to the previous 550 thousand) and we have included more difficult training data, in wetter areas (including Canada) and with more cloud cover (up to 25% as opposed to the previous 5%).

All in all, our pixel-wise false positive rate on our test set is now down to less than 0.03% (shown in Figure 2), compared with our previous 0.5%, which translates to a less than 1% false leak detection rate (shown in Figure 3, blue and green dots) with the criteria used for detecting catalogued leaks. Our main objective here was to show the quality of our model’s output compared to state-of-the-art methane retrieval fields. Other standard masking techniques such as water and landcover masks would lower our false positive rates, and is the topic of further work (along with further increasing the size of the training set).

We have added the following paragraph to the main text, that explains further experiments we have done to investigate the false positive rate of our approach:

(1.224) "Fig. 3 also summarizes our detection rate when applying the exact same methodology in the absence of known leaks, for three different tests. i) The blue bin shows our average detection rate when applying our model to pairs of Sentinel 2 images over southern New Mexico (but away from the Permian Basin), and provides an estimated false positive rate of 0.7% in conditions similar to that of the Permian Basin. ii) The green bin shows our false detection rate using the exact same methodology as elsewhere in this figure, but for pairs of Sentinel 2 images from our test set with no plume embedded. This yields a false positive rate of 0.9%, which is in agreement with a pixel-wise false positive rate estimated below 0.03% (as shown in Fig. 2)."

1(b). Lines 69-70 state their approach fully automated monitoring of emissions, but the approach produces a nontrivial amount of false negative and false positives detections which may prohibit practical use. This claim needs to be changed or better supported by the analysis in the paper.

Thank you for your comment. As described above, we were able to significantly reduce false positives rates. We also toned down some of the claims in the paper.

1(c). The authors make the claim on lines 82-84 that the database contains areas that do not encompass known potential methane sources, but they do not thoroughly describe how they ensured this while sampling, they do not supply an exhaustive list of potential methane sources, and they state they were only measured far from oil and gas activity on line 160, which I believe puts into question whether the claim on lines 82-84 is true. The sampling process should be described in more detail, and if the authors believe there could be known potential sources in the negatives, this claim should be modified and conclusions made from results should be adjusted accordingly if needed.

Thank you for your suggestion. When sampling the data, we performed a visual and landcover analysis of the areas covering the extent of each selected tile to try to ensure that there were no oil and gas facilities, landfills, natural gas plants, or other obvious sources of potential methane point sources. This is of course not entirely foolproof, but corresponds to the best that we can possibly think of in order to build a dataset relying on real Sentinel-2 data.

We have added the following paragraph to the Methods section:

(1.327) “We performed a visual inspection of the optical and landcover extent of the selected tiles to avoid, as much as possible, potential sources of point-source methane emissions”.

In order to evaluate if our training, validation and testing data are really methane free, we compare our false positive rate on our database (green and blue points in Fig. 3), to our false positive rate when applying our model in oil and gas basins at random (red point in Fig. 3). The dramatically enhanced random detection rate near oil and gas activity (4 to 5 times, from 0.7-0.9% to close to 4%) shows that methane sources are at least much rarer in our database. Because there is no ground truth with respect to methane emissions, we argue that this fits the description that there are “tentatively” no methane sources in our database.

We have added the following description to this new experiment:

(1.232) “The red bin shows our deep learning model’s detection rate over the regions surveyed in the Carbon Mapper catalogue, but at random locations and times (instead of centering the model’s input on known leaks). This detection rate essentially shows the chances of detecting a different leak (and/or making a false detection) at random when assessing the detection of a particular leak of interest in an area containing methane sources.”

2. Several methodological decisions were made without sufficient motivation and explanation, some of which may affect the results and conclusions made from the paper.

Thank you for your comment and suggestions. We have tried to add more details about the motivation of the approach in the main text, and to add explanations regarding individual operations. These changes are described for each listed point below:

2(a). Why use a single pair of consecutive times? Why only predict a single plume image? Does the plume correspond to one of the times, the difference, or something else?

The reliance on two time-steps has been chosen as an analogy to the gold standard method used to detect methane in Sentinel-2 data, the Multi-band multi-pass (MBMP) approach compared against in several figures of the paper. The MBMP method focuses on the analysis of band ratios (using bands 11 and 12) at two time-steps. This is because many of the noise sources in Sentinel-2 data are due to surface properties and land cover, and are more or less independent of the time at which they are captured. The use of two time-steps allows the model to discriminate between persistent SWIR absorption in the same exact location (likely due to noise) and transient SWIR absorption at the exact same location (which may be due to methane).

The output of the model corresponds to the plume detected in the second image. This is because (similarly as MBMP) the first image in time should be understood as a ‘reference image’, to which the second is compared to identify local absorption patterns in SWIR that are not persistent in time and space.

To improve the clarity of the description, and emphasizes its parallel with the current state-of-the-art (MBMP), we have added the following paragraph to the main text:

(1.147) “The use of two time-steps as input (conceptually similar to the MBMP approach) enables the model to use the first image as a reference image, to which the second image is compared in order to identify transient signals in methane absorbing band 12, while false positives in band 12 can be discriminated using the other bands and their evolution over the two time-steps. This comparison is crucial to distinguish signal from noise and correctly detect methane plumes, in particular the smaller ones.”

2(b). Why are all input bands sampled to 20m? How is the downsampling for 10m bands and upsampling for 60m bands done?

The input bands are sampled to 20m because this is the resolution of the SWIR bands in Sentinel-2. The particular band typically used to detect methane, B12, has a 20m resolution. Keeping this resolution leaves the most important bands of the data with respect to methane detection (bands 11 and 12) untouched. The resampling of the bands with a different resolution than 20m is done by nearest neighbor resampling, and we have added this details to the Methods section.

(1.334) “All spectral bands are re-sampled to the 20m resolution of band 12, using nearest neighbor resampling. We keep the resolution of the SWIR bands unchanged”.

2(c). What operation is used to perform the upsampling in the UNet architecture?

We have added details describing our deep learning model, including details on upsampling, which is done using deconvolutional layers (Pytorch ConvTranspose layers). We added this information in the caption of Figure 3.

2(d). Why did the authors design and use the proposed deep learning architecture rather than other well-established architectures? ViTs are known to struggle with dense prediction tasks, so comparing to well-established CNN-based segmentation architectures and ViT-based segmentation architectures that have been modified for dense prediction would much better support the use of this architecture, should it outperform those approaches. Furthermore, several implementation details were left out, like the optimization procedure (optimizer, learning rate, batch size, etc.) for example.

We have added details on the optimizer, learning rate, batch size etc. to the Methods section:

(1.371) “Our deep learning model is trained on batches of 64 samples of pairs of 128x128 Sentinel 2 tiles, in which random Gaussian plumes are embedded. The model is trained for 10 epochs using the Adam variation of stochastic gradient descent, with a learning rate that starts at 10^{-3} and is progressively reduced by 0.1% when there is no improvement in validation performance after 10 batches. The model that has the best performance on the validation set is kept.”

We have found that the main difference between our transformer auto-encoder and pure CNN auto-encoder architectures is that the transformer architecture is able to determine that methane signals downwind from a plume it has detected are more likely to be real methane signals. In other words, the transformer-based model is able to recognize long-range and anisotropic dependencies in the signal we are looking for. This is a subtle difference that is not obvious in raw performance metrics, but still noticeable, and we have added performance figures similar to Fig. 2 comparing a purely convolutional Unet-resnet architecture to the transformer-unet architecture, in Fig. S6.

2(e). Why split datasets by region if the data is synthetically generated? Wouldn't one want to include all regions in the training set to improve the generalization, rather than excluding some regions?

Thank you for your comment. We could indeed do this, but our goal here was to demonstrate that the model generalizes well. In particular, showing its performance on a test set sampled from different regions shows that it didn't simply overfit the training regions. This is a better demonstration of the robustness of the model. We note that the training data is only partially synthetic, with the background Sentinel 2 being real data in which we embed synthetic plumes, and generalizing to different Sentinel 2 backgrounds is therefore critical.

Fine-tuning the model to areas of application could be helpful when used in practice. We added the following paragraph to the main text to emphasize this point:

(1.252) "Finetuning on specific regions of interest could also be performed in order to improve performance and limit false positives when using the model in practice".

2(f). Regarding the argument made on lines 178 - 181, even though training on synthetic data was not centered around the plumes, doing so when testing the model is biased. If the goal is to understand how well the model works for detecting real methane plumes in practice, the locations of the methane plumes and often even the potential sources will not be known a priori and therefore the images will not be centered on the plumes. The evaluation needs to be redone without centering it in order to identify whether the model is robust to perturbations in the locations of the real methane plumes.

The evaluation in the test set (Figure 2) is not done using centered plumes. The plumes are only centered for the application to real airborne detections, for visualization purposes. We have added a new figure in the Supplementary Materials (Figure S5) analyzing the distribution of our model's detections on test set data, that illustrates that the model is not positively biased towards detections in the center of the image.

3. No limitations are stated even though there are several, including but likely not limited to (1) the use of randomly sampled negatives which may have methane emissions, which is likely reducing the sensitivity and leads to a rough approximation the false positive rate, (2) the approximated false positive rate is nontrivial, (3) the uncertainty of real-world detection accuracy arising from intermittency of plumes.

In the 3 months we have been working on the revised version of this paper, we have tried to address to some extent these limitations, by: i) reducing our false positive rate by further training the model on more data, increasing the amount of wet and cloudy training samples while avoiding cloudy samples when assessing against the Carbon Mapper methane leaks; ii) comparing our false leak detection rate away from oil and gas activity (0.7 to 0.9%) and near oil and gas activity (around 4%) and iii) demonstrating the performance of our model on controlled releases, that our model correctly and blindly detects.

There remain lots of uncertainty for real-world detections below 1 ton/hour (due to the absence of previously detected leaks in Sentinel 2 data below this threshold, preventing any direct comparison), and we have added caveats and discussions in this regard throughout the manuscript. Notably, we now insist on the caveat that our model's performance is dependent on plume size rather than on source rate.

Abstract (1.20) “We compare our detections with airborne methane measurement campaigns and show that our method is able to detect most methane point sources in Sentinel 2 multi-spectral satellite data down to plumes of 0.01 km², corresponding to 200 to 300 kg CH₄ h⁻¹ sources (with variations depending on wind conditions). The assessment of our model on synthetic data as well as on real methane plumes results in an order of magnitude improvement over the state-of-the-art, providing a significant step towards the automated, high resolution detection of methane emissions at a global scale, every few days.”

Introduction (1.69) “Our results suggest that our model detects most methane emissions down to plumes of 0.01 km², corresponding to methane leak rates of 200 to 300kg/h (with variations depending on wind conditions), which accounts for the vast majority of the estimated methane budget coming from point sources for most airborne campaigns in California, Colorado, and the Permian Basin analyzed in a recent survey.”

Minor Comments:

Overall

1. The syntax used throughout the paper is good. However, paragraph organization and structuring could use work. For example, why are lines 110-118 three separate paragraphs? Why are lines 218-224 separate paragraphs?

Thank you, we modified the structure of the text.

2. The authors are not consistent with spelling “multi-spectral”, sometimes excluding the hyphen. It would be good to make this consistent and look for other potential inconsistencies throughout the paper.

Thank you for this comment. We have modified the manuscript to ensure consistent spelling.

3. The Figures (specifically 1 and 3) could be substantially improved to improve readability. 3 (a). Figure 1 could more clearly show how the images are being divided into patches, which is an important design decision. It could also be more aesthetically designed overall, including using visual representations of the operations that are more standard (and in my opinion, more clear).

Thank you for your suggestion. We have modified Figure 1 for greater clarity.

3 (b). In addition to the changes I suggested in the major comments, Figure 3 could include horizontal gridlines to improve readability.

Thank you for your suggestion, we modified Figure 3 accordingly.

4. The Data availability and code availability suggest neither the curated dataset nor model training code will be made publicly available. However, the dataset and code would be fantastic resources for other researchers to be able to use and, in my opinion, releasing them would lead to the work making a much larger impact.

This work results from a public/private partnership and part of the code is proprietary. However, we will make the model available for use in the cloud, along with its detections. We note that porting the model in the cloud is an ongoing work and will also enable global automatic detection of methane, which will be the topic of future publications.

Abstract

1. The Abstract could cite more specific numbers, e.g. the capture rate.

Thank you for this suggestion. We modified the Abstract as follows:

(1.19) “We compare our detections with airborne methane measurement campaigns. Our results suggest that our method is able to detect most methane point sources in Sentinel 2 multi-spectral satellite data down to plumes of 0.01 km², corresponding to 200 to 300kg CH₄ h⁻¹ sources (with variations depending on wind conditions).”

Introduction

1. The background and motivation for the work is solid. However, the claims made in the last paragraph are exaggerated. Lines 67-70 again claim that the models robustly detect emissions down to 200kg/h, and that it unlocks fully automated monitoring, which I don't think the current results sufficiently support as I've explained above.

Thank you for your comment. We have modified the claims accordingly:

(1.73) “Our approach represents a significant step towards the automated monitoring of persistent methane emissions, anywhere on Earth, every few days, and would require few modifications to enhance detection capabilities in other multi- and hyper-spectral constellations.”

Results

1. It'd be helpful to plot the locations of the training / validation / test tiles on a map, at least as a supplementary figure, so that readers can gain a better understanding of the geographic distribution of the splits.

We added a new figure to the supplementary materials (Figure S7) to show this distribution.

2. References to Fig. 2B and 2C are swapped in the text from lines 139-157 (or the figure/caption labels need to be swapped)

Thank you for catching this! We modified the text.

3. The model is not a transformer. It is a semantic segmentation model with a vision transformer encoder. I'd suggest modifying language around this throughout the paper (e.g. lines 102, 110), potentially including the title.

Thank you for your comment. We have modified the main text accordingly.

4. Line 114 and the caption state that the model is an auto-encoder but it is not. An auto-encoder attempts to reconstruct the input. This is an encoder-decoder segmentation architecture.

We have corrected the main text accordingly.

5. Lines 158-160 claim the FPR is reliably evaluated, but I think the authors are trying to say that their estimated FPR is likely an upper bound of the true FPR, given some of the detected plumes may actually be real plumes, suggesting it is not reliably evaluated. Additionally, for the exact reason the FPR is likely overestimated, the FNR may be underestimated, which could be inflating the AUC.

Methane leaks away from oil and gas activity (or other potential sources not spotted during the visual inspection of the tiles), and that are detectable in Sentinel 2 data, are likely to be extremely rare events, that we believe we can neglect. We have removed the claim of a “reliable” estimation of the FPR, and we simply insist on the fact that the negative samples are not synthetic:

(l. 176) “Importantly, we note that the false positive rate of our model is evaluated here on real and unaltered Sentinel 2 data. Only the positive samples of our databases have a synthetic element, while the negative samples are original Sentinel 2 data.”

6. Why switch between kg/h and tons/h on line 167? Better to stay consistent with one (or report both).

We modified the text for greater consistency.

Discussion

1. I think the discussion does a better job of making more supportable conclusions than other parts of the paper. However, the claim on lines 233-236 again seems too strong as the authors do not present results which sufficiently compare to these constellations. I'd suggest softening this claim or including these results.

Thank you for your comment. We modified the claims as follow:

(l.294) “A timely global monitoring system for methane, able to detect emissions down to a few hundreds of kilos per hour, would be a fundamental stepping stone towards an automated, complementary and holistic monitoring system to building inventories of anthropogenic emissions at scale. In particular, such emissions account for the vast majority of U.S. methane point-sources in volume from methane-emitting areas analyzed in a recent survey of airborne campaigns in several States.

By drastically lowering detection capabilities in multi-spectral data, our results suggest that public, general-purpose multi-spectral satellites can be turned into powerful methane monitoring tools capable of reaching detection performances approaching that of hyper-spectral constellations built specifically for methane detection, with the potential of generating global methane inventories at fine spatial and temporal scales.”

Methods

1. How were the Sentinel-2 images downloaded?

The data was downloaded using the PEPS API (a mirror of ESA's repository). We added this information to the Methods section.

2. How was cloud cover determined?

The cloud cover was determined using the metadata provided by ESA for each Sentinel-2 image. We added this information to the Methods section.

References

- [1] Sherwin, E. D. *et al.*, Single-blind test of nine methane-sensing satellite systems from three continents. *EarthArXiv* (2023)
- [2] Cusworth, D. H. *et al.*, Intermittency of Large Methane Emitters in the Permian Basin, *Environ. Sci. Technol. Lett.*, 8, 7, 567–573 (2021)
- [3] Ehret, T. *et al.* Global tracking and quantification of oil and gas methane emissions from recurrent sentinel-2 imagery. *Environmental Science & Technology* 56, 10517–10529 (2022)
- [4] Lauvaux, T. *et al.* Global assessment of oil and gas methane ultra-emitters. *Science* 375, 557–561 (2022).

Reviewer #1 (Remarks to the Author):

The authors have significantly improved their paper in addressing the reviewer remarks. Figures 3 and 4 and the related discussion are much more clearly presented, and the model's application to real controlled release plumes provides solid evidence of its detection skill. I appreciate the added explanations of why the model is successful and the softening of claims throughout the manuscript with respect to automatic detection of methane point sources down to 200-300 kg/h.

Below are a few more suggestions to improve the paper prior to publication:

- I think it is important to state clearly that you have **proven** or **demonstrated** fully automated detection of a point source down to 1100 kg/h (with the controlled release example) and **inferred** a detection capability down to 200-300 kg/h (from the Carbon Mapper comparison). I think this conclusion should appear at least in the Discussion section, if not also in the Abstract. As far as I can tell, currently the distinction is made only indirectly in the Results section on page 11 (lines 270-275).
- Abstract line 21: Adding to the above recommendation, I would suggest removing the word "most" here. The statistical argument around the indirect comparison with Carbon Mapper plumes is compelling, but without 1-1 verification against real plumes of magnitude 200-300 kg/h, I don't think one should state even a generic proportion of successful detections ("most"). The analysis does suggest that the model can detect plumes down to those relatively low rates, which is an important accomplishment, but as mentioned above – it's really an inferred detection capability.
- Introduction line 69: Like the previous points, I would suggest a rephrasing here. Something along the lines of "inferring a detection capability" down to 200-300 kg/h from the comparison with Carbon Mapper.
- The work is now appropriately described as a "fundamental steppingstone" towards systematic global detection and monitoring of methane point sources down to a few hundred kg/h. It would be helpful then to say a little about what comes next. What work still needs to be done to achieve "the precise, systematic monitoring of methane emissions, anywhere on Earth, every few days" (quoting from the Discussion section)? Extending the work to emission quantification, validating it against more co-detections and/or controlled releases, testing its efficacy in different observing conditions, etc. – these are just some suggestions. My point here is that it would be helpful to provide some context for future work in the Discussion section (could be just a few sentences).
- Throughout: Unlike NASA/USGS Landsat, the ESA Sentinel missions take hyphens, e.g., "Sentinel-2" rather than "Sentinel 2".
- I just noticed the hyphen in the title, "multi-spectral". This should be removed. "Multispectral" is the correct term.
- Code availability: I am not sure what Nature Communications requires but it would be highly desirable to publish the model/weights along with the final manuscript.

Well done!

Reviewer #1 (Remarks on code availability):

The code has not been made available. The authors state that they will make their model available shortly after publication. I believe the model should be made available with DOI included in the final publication.

Reviewer #2 (Remarks to the Author):

The authors have addressed the vast majority of my concerns. I commend the authors for all their work addressing many of the issues and limitations, and I do think the manuscript has improved substantially because of their work.

I just have a few remaining concerns which I have listed below.

1. The explanations for the strength of the model on lines 142-154 are helpful, but these are testable claims that experiments could support:

- (i) An "easy" way to test this would be to train on a random sample of the synthetic data that is similar in size to previous works (or multiple sizes to clearly show the effect of training set size). Even better would be to train the model on the data used in prior works.
- (ii) The impact of two time-steps could be tested by removing the reference image.
- (iii) This has now been tested, although I think it would be even better to compare to other ViT-based architectures as I explain below.

2. Although I appreciate the authors comparing to a CNN-based architecture, as I stated in my initial review, comparison to other more well-established ViT-based segmentation architectures (e.g. SegVit [1]) seems necessary as well given the authors are proposing a custom architecture. Moreover, using a custom one prevents the use of pre-trained weights for the full architecture, which may negatively affect performance, so comparing to a pre-trained SegVit would be informative.

(a) On the last point, a potentially important detail is missing from the paper: did the authors randomly initialize the network, or were any parts of it initialized with pre-trained weights (e.g. the encoder)?

3. Other important methodological details that should be included:

(a) What size vision transformer was used (tiny, small, base, large, huge)?

(b) What's the patch size?

(c) Were other architecture sizes or patch sizes explored, and the chosen settings found to perform the best?

4. The current title is slightly incorrect, as only one transformer was trained and the whole model is not a transformer. One suggestion would be to change it to "Automatic Detection of Methane Emissions in Multi-Spectral Satellite Imagery Using a Vision Transformer".

[1] SegViT: Semantic Segmentation with Plain Vision Transformers. NeurIPS 2022.

Once again, all our thanks for the time and effort spent by the reviewers on our paper. We sincerely believe that our manuscript has been made much stronger after following the advice of the reviewers over the two rounds of revisions, and we are extremely grateful.

In the following, the reviewers' comments are in black and our answers are in light blue, and the edited text in the manuscript is in dark blue.

REVIEWERS' COMMENTS

Reviewer #1 (Remarks to the Author):

The authors have significantly improved their paper in addressing the reviewer remarks. Figures 3 and 4 and the related discussion are much more clearly presented, and the model's application to real controlled release plumes provides solid evidence of its detection skill. I appreciate the added explanations of why the model is successful and the softening of claims throughout the manuscript with respect to automatic detection of methane point sources down to 200-300 kg/h.

Below are a few more suggestions to improve the paper prior to publication:

- I think it is important to state clearly that you have **proven** or **demonstrated** fully automated detection of a point source down to 1100 kg/h (with the controlled release example) and **inferred** a detection capability down to 200-300 kg/h (from the Carbon Mapper comparison). I think this conclusion should appear at least in the Discussion section, if not also in the Abstract. As far as I can tell, currently the distinction is made only indirectly in the Results section on page 11 (lines 270-275).

We have added text to the introduction that clarifies that we directly prove our detection capabilities down to 1100 kg/h, and indirectly demonstrate them down to 200-300 kg/h:

“Our results also show that our model can detect all the methane releases that have been timed with Sentinel-2 overpasses down to 1100kg/h, with a gap in S2 controlled releases below that threshold that we hope will be filled in future tests of our method” ll. 64-67

- Abstract line 21: Adding to the above recommendation, I would suggest removing the word “most” here. The statistical argument around the indirect comparison with Carbon Mapper plumes is compelling, but without 1-1 verification against real plumes of magnitude 200-300 kg/h, I don't think one should state even a generic proportion of successful detections (“most”). The analysis does suggest that the model can detect plumes down to those relatively low rates, which is an important accomplishment, but as mentioned above – it's really an inferred detection capability.
- Introduction line 69: Like the previous points, I would suggest a rephrasing here. Something along the lines of “inferring a detection capability” down to 200-300 kg/h from the comparison with Carbon Mapper.

We have removed the mention of “most” from the abstract and from the introduction.

- The work is now appropriately described as a “fundamental steppingstone” towards systematic global detection and monitoring of methane point sources down to a few hundred kg/h. It would be helpful then to say a little about what comes next. What work still needs to be done to achieve “the precise, systematic monitoring of methane emissions, anywhere on Earth, every few days” (quoting from the Discussion section)? Extending the work to emission quantification, validating it against

more co-detections and/or controlled releases, testing its efficacy in different observing conditions, etc. – these are just some suggestions. My point here is that it would be helpful to provide some context for future work in the Discussion section (could be just a few sentences).

Thank you for your recommendation, we have added the following “future work” paragraph to the discussion section:

“Future developments of our method will include further reducing the remaining false positives, notably by building ensembles of models and incorporating auxiliary data (e.g. land cover, wind, soil moisture, overall atmospheric methane concentration, etc.), and combining our methane detections with source rate quantification methods. In order to scale-up towards global detection of point sources of methane, we will implement our entire pre-processing and deep detection in the cloud. Last but not least, future work will also include collaborating to assess our method on methane controlled release experiments timed with Sentinel-2 and below 1100 kg/h.” ll.308-315

- Throughout: Unlike NASA/USGS Landsat, the ESA Sentinel missions take hyphens, e.g., “Sentinel-2” rather than “Sentinel 2”. Thank you for noticing, we have made the change.
- I just noticed the hyphen in the title, “multi-spectral”. This should be removed. “Multispectral” is the correct term. Thank you for noticing, the title has changed accordingly.
- Code availability: I am not sure what Nature Communications requires but it would be highly desirable to publish the model/weights along with the final manuscript. The model will be made available soon after publication in the cloud.

Well done!

Many thanks for your encouragement and for the time spent on helping us improve the manuscript!

Reviewer #1 (Remarks on code availability):

The code has not been made available. The authors state that they will make their model available shortly after publication. I believe the model should be made available with DOI included in the final publication.

We have created a CodeOcean capsule that enables to reproduce the figures of the manuscript, and we will strive to make the model available in the cloud for research purposes.

Reviewer #2 (Remarks to the Author):

The authors have addressed the vast majority of my concerns. I commend the authors for all their work addressing many of the issues and limitations, and I do think the manuscript has improved substantially because of their work.

I just have a few remaining concerns which I have listed below.

1. The explanations for the strength of the model on lines 142-154 are helpful, but these are testable claims that experiments could support:

- (i) An “easy” way to test this would be to train on a random sample of the synthetic data that is similar in size to previous works (or multiple sizes to clearly show the effect of training set size). Even better would be to train the model on the data used in prior works.
- (ii) The impact of two time-steps could be tested by removing the reference image.
- (iii) This has now been tested, although I think it would be even better to compare to other ViT-based architectures as I explain below.

The additional tests proposed here by reviewer 2 require to train from scratch other models, and is beyond the scope of our manuscript, which was aimed at demonstrating the power of deep learning

models to detect methane in Sentinel 2 data. The goal of this manuscript was not to find the best possible model architecture to do so, which will be to topic of further research and publications.

2. Although I appreciate the authors comparing to a CNN-based architecture, as I stated in my initial review, comparison to other more well-established ViT-based segmentation architectures (e.g. SegVit [1]) seems necessary as well given the authors are proposing a custom architecture. Moreover, using a custom one prevents the use of pre-trained weights for the full architecture, which may negatively affect performance, so comparing to a pre-trained SegVit would be informative.

(a) On the last point, a potentially important detail is missing from the paper: did the authors randomly initialize the network, or were any parts of it initialized with pre-trained weights (e.g. the encoder)?

The model is trained entirely from scratch, which we now specify in the Methods section, line 372.

3. Other important methodological details that should be included:

(a) What size vision transformer was used (tiny, small, base, large, huge)?

(b) What's the patch size?

(c) Were other architecture sizes or patch sizes explored, and the chosen settings found to perform the best?

We have added these details of the architecture to the Methods section as well:

“Our deep learning model has a U-net architecture, with its encoder portion replaced by a ViT, and its decoder left as a convolutional layers, with matching up-sampling using deconvolutional layers. The ViT portion of our model is the base variant introduced in the original paper, with a patch size of 16, which we train from scratch along with the decoder portion of the model.” ll. 369-375

4. The current title is slightly incorrect, as only one transformer was trained and the whole model is not a transformer. One suggestion would be to change it to “Automatic Detection of Methane Emissions in Multi-Spectral Satellite Imagery Using a Vision Transformer”.

Thank you for your suggestion, we have indeed changed the title of the manuscript.

[1] SegViT: Semantic Segmentation with Plain Vision Transformers. NeurIPS 2022.